# Neuromotor prosthetic to treat stroke-related paresis: N-of-1 trial

Mijail D. Serruya [1✉], Alessandro Napoli[1], Nicholas Satterthwaite[1], Joe Kardine [1], Joseph McCoy[2], Namrata Grampurohit [3], Kiran Talekar[4], Devon M. Middleton[4], Feroze Mohamed[4], Michael Kogan[5], Ashwini Sharan[5], Chengyuan Wu [5] & Robert H. Rosenwasser[5]

## Abstract

**Background** Functional recovery of arm movement typically plateaus following a stroke, leaving chronic motor deficits. Brain-computer interfaces (BCI) may be a potential treatment for post-stroke deficits

**Methods** In this n-of-1 trial (NCT03913286), a person with chronic subcortical stroke with upper-limb motor impairment used a powered elbow-wrist-hand orthosis that opened and closed the affected hand using cortical activity, recorded from a percutaneous BCI comprised of four microelectrode arrays implanted in the ipsilesional precentral gyrus, based on decoding of spiking patterns and high frequency field potentials generated by imagined hand movements. The system was evaluated in a home setting for 12 weeks

**Results** Robust single unit activity, modulating with attempted or imagined movement, was present throughout the precentral gyrus. The participant acquired voluntary control over a hand-orthosis, achieving 10 points on the Action Research Arm Test using the BCI, compared to 0 without any device, and 5 using myoelectric control. Strength, spasticity, the Fugl-Meyer scores improved.

**Conclusions** We demonstrate in a human being that ensembles of individual neurons in the cortex overlying a chronic supratentorial, subcortical stroke remain active and engaged in motor representation and planning and can be used to electrically bypass the stroke and promote limb function. The participant's ability to rapidly acquire control over otherwise paralyzed hand opening, more than 18 months after a stroke, may justify development of a fully implanted movement restoration system to expand the utility of fully implantable BCI to a clinical population that numbers in the tens of millions worldwide.

## Plain language summary

Stroke is a restriction of blood flow to part of the brain and can lead to chronic issues with a person's ability to control the limbs. The aim of this study was to see if a new type of device could restore movement in a person with arm weakness due to a stroke that occurred a year earlier. In our trial, a sensor was implanted into the surface of the brain, near the site of the stroke, and was connected to a computer that generated a command to open and close the hand with a motorized brace worn on the hand. This person was able to use their own brain activity to trigger the brace and pick up and move objects. This research could support the development of similar medical devices to restore movement in people who have had strokes.

[1] Center for Neurorestoration, Thomas Jefferson University, Philadelphia, PA 19107, USA. [2] Jefferson Rehabilitation, Thomas Jefferson University, Philadelphia, PA 19107, USA. [3] College of Rehabilitation Science, Thomas Jefferson University, Philadelphia, PA 19107, USA. [4] Department of Radiology, Thomas Jefferson University, Philadelphia, PA 19107, USA. [5] Department of Neurosurgery, Thomas Jefferson University, Philadelphia, PA 19107, USA. ✉email: Mijail.Serruya@jefferson.edu

Stroke is a leading cause of disability[1] with a global prevalence of over 42 million people in 2015[2], affecting over four million adults in the United States alone with 800,000 new cases per year[3]. Stroke leads to permanent motor disabilities in 80% of cases[4], and half of stroke survivors require long term care. Brain computer interface (BCI) technologies offer a potential solution to restore functional independence and improve health in people living with its effects. In the past decade, intracortical BCI technology has continued to advance, with multiple groups demonstrating the safety and efficacy of this approach to derive control signals[5–7] to restore communication and control. In parallel, wearable robotic orthosis technology can benefit patients with weakened limbs[8,9]. This single-patient pilot clinical trial sought to prove that a commercially available powered arm orthosis could be linked to the cerebral cortex in an adult with the most common form of chronic stroke. A direct path from the brain's motor centers to the orthotic could reanimate a paralyzed limb to enable useful hand and arm function.

Several signal sources have been coopted to provide commands to move paralyzed limbs. Electromyographic (EMG) control of a powered orthosis or functional electrical stimulation (FES) of muscles, has proven problematic either because users could not generate sufficient or reliable activity to provide a good control signal, or because voluntary activation of those recorded muscles (that were intended to generate the command) was opposed by the stimulator's effects[10]. Contralaterally controlled electrical stimulation- where activity from the unaffected arm triggers stimulation on the paretic arm- is a useful therapeutic intervention to improve function in the weaker limb[11] but it is not clear how this unnatural command source could be generalized to continuously-worn devices that enable independent arm movements. Several groups have explored scalp EEG, which is closer to the command's origins, to derive control signals to drive robotic braces, and in one case, FES[8,10,12,13]. While using EEG-derived signals may be promising for rehabilitation therapy, it would not be feasible for daily independent function because skin sweat and hair can cause impedances to fluctuate, compromising signal quality. Daily application of even a subset of contacts to the same skin sites can lead to skin breakdown and cellulitis. Further, EEG signals are limited in the commands that can be easily and reliably derived from the available signal. By contrast, intracortical interfaces offer a rich source of high resolution, multidimensional control signals, since it is the origin of such signals in healthy adults, in non-human primates and in people with spinal or brainstem disorders[7,14,15].

While most strokes involve cerebral white matter and direct parenchymal damage, intracortical neuromotor prosthetics have not been tested in people with strokes above the mesencephalon. It is not known whether motor cortex remains a reliable signal source in this large population. A proof-of-concept that a brain-computer interface, based on micro-electrode arrays implanted in intact cortex above a subcortical stroke, could restore behaviorally useful independent, voluntary movement, could lead to the development of a fully implantable medical device that, in principle, could reverse the motor deficits caused by stroke. The purpose of this study was to show whether an assistive brain-computer interface, when in use, could provide a behaviorally useful benefit in motor function.

Here we found that neural activity recorded from the cerebral cortex overlying a chronic, subcortical stroke, generated activity patterns related to both actual and imagined movements in the contralateral, weakened limb of a person. The recorded neural activity was decoded in real-time to control a motor on a brace worn on the weakened arm, and the participant was able to use this decoded signal to voluntarily open and close the hand to perform tasks in a home setting.

## Methods

Approval for this study was granted by the US Food and Drug Administration (Investigational Device Exemption) and the Thomas Jefferson University Institutional Review Board (protocol number 17D.459). The participant described in this report has provided permission for photographs, videos and portions of his protected health information to be published for scientific and educational purposes. After completion of informed consent on March 4, 2020, medical and surgical screening procedures, two MultiPorts (Blackrock Microsystems, UT), each comprising two $8 \times 8$ platinum tipped microelectrode arrays tethered to a titanium pedestal connector, were implanted into the cortex of the precentral gyrus using a pneumatic insertion technique[16,17]. Details of the human surgical procedure are in preparation for publication and followed other similar studies. Trial selection criteria are available online (see Clinicaltrials.gov, NCT03913286 and Supplementary Table 1). The trial was designed with the implantation phase to last a maximum of three months (Fig. 1).

**Participant.** The participant was a right-handed male who experienced right hemispheric stroke, manifest as acute onset dense left hemiparesis and expressive aphasia, at which time he was age between ages 35 and 40. Due to unknown time of onset and hypertension at presentation, the participant was not a candidate for thrombolysis. CT angiogram showed occlusion of the right posterior cerebral artery and high-grade stenosis of the left posterior cerebral artery in the proximal P2 segment. MRI of the brain showed acute infarcts in the right basal ganglia/corona radiata and right occipital lobe. He was started on dual anti-platelet therapy for 3 weeks and then was transitioned to aspirin 81 mg once daily, along with atorvastatin and anti-hypertensives. He had left-sided hemiparesis, dysphagia, left homonymous hemianopia and dense left visual neglect and was transferred for inpatient rehabilitation. Over a period of three months, aphasia and dysphagia resolved and he learned to ambulate independently, albeit with a persistent left foot drop. Neuroimaging showed evidence of multi-focal strokes, and prior silent strokes. The participant had previously been in good health and did not

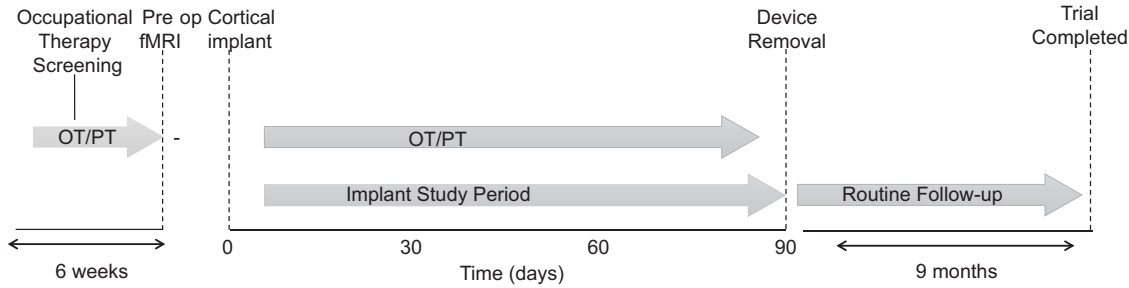

**Fig. 1 The clinical trial timeline.** This figure shows the sequence of the Cortimo clinical trial.

have any known stroke risk factors such as diabetes or smoking. There was a history of loud snoring and the participant had not been evaluated for obstructive sleep apnea. Transthoracic and transesophageal echocardiography were normal as were serial hypercoagulability panels; the participant was adopted and the biological family history unknown. The participant was deemed to have had embolic strokes of unknown source. Although serial electrocardiography since the stroke was normal, the participant is being scheduled for a loop recorder to survey for possible paroxysmal atrial fibrillation. The participant had learning disabilities and was presumed to have had mild cognitive impairment prior to the stroke. Screening formal neuropsychological testing identified neurocognitive problems (full scale IQ 59) and concluded that the participant remained fully capable to provide proper informed consent and to participate in this trial, meeting its demands and requirements. The participant provided both verbal and written informed consent, both to participate in the trial and to share his identifying information with the public. He had been working full time at the time of the stroke and has been unable to return to work since the stroke. Clinical assessment of the participant included neurological exams one year, six months and one month prior to enrollment in the setting of routine outpatient care; the exam was serially repeated once enrolled. Clinical assessment included detailed history review and confirmation of meeting all selection criteria. The trial was constructed to include a six-week screening phase (Fig. 1), during which the participant underwent occupational therapy (1 h per session, three times per week) to assess how well the participant could understand and master use of the MyoPro device. The neuropsychological testing also took place during the screening phase. Predefined outcome measures (described subsequently) were also recorded during the screening phase. During this phase, therapy consisted of evaluation of active and passive range of motion, strength, hypertonicity, and goal setting with ADLs and IADLs. Treatment of left upper extremity spasticity was completed utilizing stretching, functional electrical stimulation, and creating a splinting schedule. This pre-implant occupational therapy was considered screening in that it was decided prior to enrolling the participant that proceeding to implant would only occur if the occupational therapist felt that the participant could adequately comply with the therapy. Therapeutic exercise and activity were incorporated to improve postural control and non-volitional movements with left upper extremity. Introductory use of MyoPro device was incorporated within treatment. Pre-implant physical therapy included baseline functional balance measures with interventions focused on open/closed chain strengthening, static/anticipatory/dynamic postural control and gait training.

**Pre-operative fMRI**. The participant underwent MRI on a 3 T Philips Ingenia MRI scanner. A 1 mm isotropic 3-D T2 FLAIR was obtained for structural localization. A single shot echoplanar gradient echo imaging sequence with 80 volumes, repetition time (TR) = 2 s, echo time (TE) = 25 msec, voxel size = 3 × 3 mm2, slice thickness = 3 mm, axial slices = 37. The participant was asked to visualize movements of his paretic left hand during the MRI. Each motor trial consisted of a block design featuring a 20 s rest block and a 20 s active block repeated. This block design was repeated between 4 times for a total of 240 s scans. Visual stimuli comprised a 20 s video depicting a 3D modeled limb at rest, followed by a 20 s video of the limb performing the desired task. Motor tasks included repeated hand open/clench or arm extension elbow and were either active (participant performed or attempted to perform motion) or passive (physician manually moved participant's arm). In active tasks, the participant was

instructed to follow the movements in the video or concentrate on following for the paretic limb. Task prioritization was based on pre-exam training of the participant's capabilities and examination of BOLD activation observed during the scan. Post processing including motion correction, smoothing, and general linear model estimation performed using SPM software (www.fil.ion.ucl.ac.uk/spm) and Nordic brain EX software (NordicNeuroLab, Bergen, Norway). Statistical maps were overlaid on the 3D T2 FLAIR image for visualization of activation.

**Cortimo system**. 'Cortimo' is the designation provided to the FDA to represent the overall system (Fig. 2) that comprised two percutaneous Multiports (Blackrock Microsystems), each in turn having two multi-electrode array sensors, the cabling, amplifiers, software and the powered MyoPro orthosis. Each sensor is an 8 × 8 array of silicon microelectrodes that protrude 1.5 mm from a 3.3 × 3.3-mm platform. At manufacture, electrodes had an impedance ranging between 70 KOhm and 340 KOhm. The arrays were implanted onto the surface of the MI arm/hand region guided by the pre-operative fMRI; with electrodes penetrating the cortex to attempt to record neurons in layer V. Recorded electrical signals pass externally through a Ti percutaneous connector secured to the skull. Cabling attached to the connector during recording sessions routes signals to external amplifiers and a computer that process the signals and convert them into different outputs, such as servo motor position of the MyoPro brace or screen position of a neural cursor. Currently, this system must be set up and managed by an experienced technician.

**MyoPro brace**. The MyoPro (Myomo, Inc, Cambridge, MA) is an FDA-cleared myoelectric powered arm orthosis designed to support a paretic arm[9]. The rigid brace incorporates metal contacts attached to soft straps that can be adjusted such that contacts rest on the biceps and triceps proximally, and on wrist flexors and extensors distally, on the paretic upper extremity. The sensors continuously record the root mean square of underlying muscle activity. Thresholds are manually set such that signals exceeding them will trigger one of the MyoPro motors. Because the participant retained residual elbow flexion and extension strength, the motor at the elbow was set up such that biceps activation triggered elbow flexion and triceps activation triggered elbow extension. For hand opening, the MyoPro was set up to either use myoelectric control, or to use BCI-based control. Since the participant was unable to voluntarily extend the wrist or open

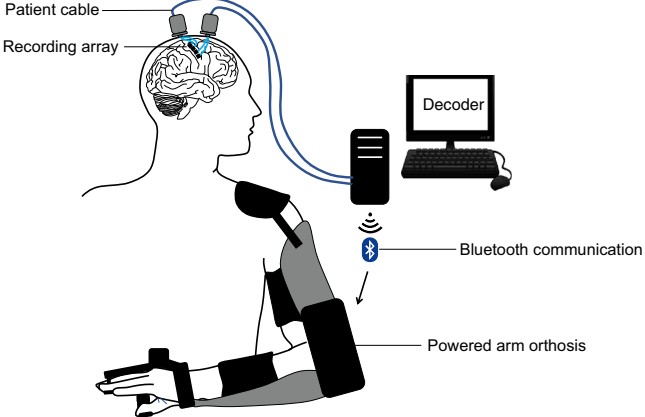

**Fig. 2 The cortimo system.** The overall device comprises two Multiports, each with two 8 × 8 microelectrode arrays, patient cables linked to external amplifiers, a decoding computer and a wearable, powered arm orthosis.

the fingers, the myoelectric mode was set up such that the default state was with the hand open, and it would only be closed by activating enough wrist flexor activity.

**MyoPro EMG control**. The MyoPro device is designed to use four EMG channels to achieve control user's control over two electric motors that control hand and elbow motion. The four EMG channels are fixed and they are: (1) forearm extensors; (2) forearm flexors; (3) triceps; and, (4) biceps. It is important noticing that the MyoPro does not use raw EMG signals for hand/elbow control mechanisms, but it calculates in real-time the rectified RMS of the raw EMG signals and the RMS signals are eventually compared against the user selected thresholds for motion classification.

Depending on the control mode selected, a combination of the above channels can be used to control either a single joint at a time or both. Namely, the available modes are dual mode, open mode or close mode. In dual mode, hand motion controlled by two manually set thresholds based on forearm EMGs. Elbow motion controlled by two different manually set thresholds based on biceps' and triceps' EMGs. The dual mode control principle is as described in the eq. below:

$$Motion\ Type = \begin{cases} Open, |Flexors_t < FlexTh\ AND\ Extensors_t > ExtTh \\ Hold, |(Flexors_t < FlexTh\ OR\ Extensors_t < ExtTh) \\ \qquad OR \\ (Flexors_t > FlexTh\ OR\ Extensors_t > ExtTh) \\ Close, |Flexors_t > FlexTh\ AND\ Extensors_t < ExtTh \end{cases}$$

(1)

MyoPro DUAL Mode Motion Control Strategy: In DUAL mode the MyoPro uses flexors and extensors for each of the two joints (hand and elbow) and two thresholds for motion control decision making.

In open mode, hand motion controlled by one manually set threshold based on forearm extensors. Elbow motion controlled by one different manually set threshold based on triceps EMGs. The open mode control principle is as described in the eq. below:

$$Motion\ Type = \begin{cases} Open, Extensors_t > ExtTh \\ Close, Extensors_t < ExtTh \end{cases}$$

(2)

MyoPro OPEN Mode Motion Control Strategy: In OPEN mode the MyoPro moves the joint into full flexion if the extensors are below the extension threshold while the joint is moved into full extension if extensors are above the extensor threshold.

In close mode, hand motion controlled by one manually set threshold based on forearm flexors. Elbow motion controlled by one different manually set threshold based on biceps. The close mode control principle is as described in the eq. below:

$$Motion\ Type = \begin{cases} Close, Flexors_t > FlexTh \\ Open, Flexors_t < FlexTh \end{cases}$$

(3)

MyoPro CLOSE Mode Motion Control Strategy: In CLOSE mode the MyoPro moves the joint into full extension if the flexors are below the flexion threshold while the joint is moved into full flexion if extensors are above the flexion threshold. where Flexors and Extensors are the corresponding EMG signals and ExtTh and FlexTh are respectively manually set thresholds for extensors and flexors.

**Pre-specified outcome measures**. More than a year prior to enrolling the participant, the trial was designed to assess specific outcome measures one month prior to device implantation, and again three months post-implantation. These metrics included the Fugl-Meyer Motor Impairment Score[18], the Action Research Arm Test (ARAT)[19], the Motricity Index[20], the Hand and Recovery Scales within the Stroke Impact Scale[21]. Although the Giving Them A Hand scale[22] was initially included as a metric on the original IDE and IRB filings, the occupational therapist co-investigators who subsequently joined the team felt that this was not a well-validated measure and it was decided, prior to enrolling the participant, that it would not be used. During the implant phase, a component of the Jebsen-Taylor measure (picking up, moving and putting down five objects, one at a time, one after the other)[23] was added because it was found that the participant was able to perform this task more consistently and easily with the orthosis than the ARAT, inspiring greater participant motivation and engagement and hence facilitating comparison of myoelectric vs BCI control modes. The outcome measures were performed by clinicians who were trained and standardized in administration, and each measure was assigned to specific co-investigators to perform serially to minimize inter-rater variability across time.

**Recording sessions**. Research sessions were scheduled five days per week at a temporary residence, adjacent to the hospital, provided to the participant. Sessions could be canceled or ended early at the participant's request. Sessions would commence with neural recording and spike discrimination. While initial sessions included filter building and structured clinical endpoint (cursor control) trials, in the final month of the trial, training-less algorithms were used with the participant proceeding directly to BCI-controlled hand action once patient cables were connected. Performance of computer tasks, orthosis control and occupational therapy exercises followed. The electrodes and neural signals selected immediately before filter building remained constant for any given session's orthosis control trials.

**Decoder filter building**. Units were extracted using an automatic thresholding approach based for each electrode channel, based on Root Mean Squared multipliers[15]. For each session, single and multiunit data or high frequency (100–1000 Hz) local field potentials derived from multiple channels (20–30) were used to create a linear filter to convert these real-time multidimensional neural features into either a one or a two-dimensional (position or velocity) output signal. Motor activity and motor imagery approaches were tested for filter building, including imagining opening and closing the paretic hand, passively flexing and extending the elbow, passively opening and closing the hand, and observing a computer cursor displayed on a monitor moving up and down without any specific instruction. Training data for building the linear filter were collected with the participant gazing at a screen where a target cursor was moved slowly up and down for one minute (5 s to go from the top to the bottom of the screen or vice versa, at 20° visual angle). After this preliminary filter was built (see next section for more details), a new 1-minute re-training session was performed, this time the manually controlled target cursor was accompanied by a prediction cursor that was neurally controlled by the participant. Using this additional training set, a second filter was built and then tested on a simple target acquisition game in which the y-position of the predicted output was discretized into zones such that positions on the upper part of the screen would cause an animation sprite to move up by a fixed distance (1 cm), and positions on the lower part of the screen would cause the sprite to move down by the same fixed distance.

**Decoder design**. The Cortimo system has been designed to provide a series of real-time decoding methods. Namely, two types of decoders have been implemented both discrete and continuous (i.e., filters). The available discrete classifiers were

Linear Discriminant Analysis (LDA), Decision Trees, Support Vector Machine (SVM), k-nearest neighbors (KNN); while the available continuous decoders were: a position-velocity Kalman filter[24] and a linear filter[25]. All decoders could be quickly trained using data and labels recorded during training sessions (filter building sessions). The Cortimo participant and the BCI system achieved the best brace control performance when the linear filter was used and the training sessions were guided by a one or a two-dimensional cursor control task. Briefly, a cursor was displayed on the computer screen and the cursor position was controlled using a weighted linear combination of (ad-hoc) selected features derived from real-time participant's brain activity. The neural features mostly used in this trial were either the cumulative (across all selected channels) spike count binned in 200 ms windows or the cumulative (across all selected channels) spectral density of Local Field Potentials (LFPs) calculated in the frequency range 100–500 Hz with 50 Hz frequency steps.

To achieve better and more reliable brace control in closed-loop tasks, the continuous (linear) filter output, originally corresponding to a specific screen location, was fed into a discretization block where the decision-making rules were either:

$$PBM_t = \begin{cases} Extension, x_t < AT \\ Flexion, x_t \geq AT \end{cases} \quad (4)$$

The Cortimo Discretization Block: Discretization decision rule number 1. where $PBM_t$ represents the hand motion at time step $t$, $x_t$ is the linear filter output at time step $t$ and $AT$ is a position threshold chosen to maximize user's control.

$$PBM_t = \begin{cases} Extension, x_t < AT_1 \\ Hold, AT_2 < x_t \geq AT_1 \\ Flexion, x_t \geq AT_2 \end{cases} \quad (5)$$

The Cortimo Discretization Block: Discretization decision rule number 2. where $PBM_t$ represents the hand motion at time step $t$, $x_t$ is the linear filter output at time step $t$ and $AT$ and $AT_2$ are respectively a lower and higher position thresholds chosen to maximize user's control.

**BCI orthosis use**. The discrete output was then used to control the aperture of the hand via the MyoPro's hand brace motor. The up-down mapping on the screen was translated into closed-open positions of the hand. The participant then performed a series of functional tasks including grasping and then dropping an object, the Action Research Arm Test[19], and a variation on Jebsen Taylor item moving test[23]. These were tested with both the participant seated and standing.

**Training-less mapping**. When the participant would attempt to overpower the orthosis motors with residual finger flexion strength, a 'training-less' approach was invented and deployed in which a rolling 1-second baseline of the LFPs signals was used to calculate spectral power in the high gamma band (100–500 Hz). Namely, 1-second long LFP continuous voltages were used for computing the average spectral density estimation in the frequency band 100–500 Hz, using non-overlapping frequency bins with a 50 Hz width. Spectral density was computed using the Matlab periodogram method. Values were updated every 500 ms, using 1-second-long rolling windows with 50% overlap. Real-time spectral features derived from the 20 most neuromodulated channels were averaged across channels to produce a single high gamma band value for each 500 ms software update. Orthosis hand-closure would be triggered by an increase in this mean spectral power from the resting baseline ranging between 0.5 and 3 $V^2/Hz$ to values greater than 10 $V^2/Hz$, where real-time values above this threshold would make the hand motor close.

**Concomitant occupational and physical therapy**. Since being discharged from acute rehabilitation 60 days after the initial stroke, the participant enrolled in outpatient physical and occupational therapy. Prior to the device implantation, the participant completed a six-week course of occupational therapy screening phase. Following device implantation, the participant continued occupational therapy, twice per week, and physical therapy, once per week, each session lasting approximately one hour. In the three-month implantation phase, the participant hence received 24 one-hour sessions of occupational therapy and 12 one-hour sessions of physical therapy. In addition, clinical trial assistants practiced therapy exercises with the participant and accompanied him to a gym for aerobic conditioning (either stationary bicycling or NuStep combined arm and foot cycle, for 20 min to a target heart rate of 120 beats per minutes): these sessions were approximately one hour and were practiced daily, including weekends for a total of 91 days. Occupational therapy focused on postural training while seated and walking, donning, and doffing the MyoPro, repetitive trials of hand open/close elbow flex/extend wth the MyoPro, and using the MyoPro for functional activities. Timed functional electrical stimulation[26] (e.g., pincer grasp programs using the XCite brand FES unit from Restorative Therapies) and vibration therapy (5 to 10 min of focal muscle vibration) were used for spasticity management[27]. Physical therapy exercises included scapular mobilization, progressive range of motion, weight bearing, forced use with game-related activities to encourage left UE volitional control, and aerobic endurance exercise. The exact exercises performed (passive and active range of motion stretching, neuromuscular education, electrical stimulation, orthosis use), blood pressure, and subjective pain reports, were logged for every rehabilitation session; in addition, clips of several sessions were recorded by video (see Supplementary Occupational Therapy Log and Supplementary Videos 1–12 in particular Supplementary Video 12).

**Reporting summary**. Further information on research design is available in the Nature Research Reporting Summary linked to this article.

## Results

The participant underwent intracortical implantation in autumn of 2020 and explantation three months later on January 2021, in accordance with the intended 3-month duration of the trial. Over the course of the study, the participant had three minor, and one serious, device-related adverse events, all of which were treated, resolved, and reported to the governing regulatory bodies. The serious adverse event was the development of a scalp infection at the left pedestal site one week prior to the device removal date, despite a regimen of topical antibiotics and regular cleaning. This infection was anticipated and was described as a potential risk in the informed consent form and consent interview. The left pedestal site had posed a challenge since the time of the initial surgery as it was not possible to exactly re-approximate the skin flap leaving the base of the pedestal exposed. This area was protected and granulated and grew new skin. The participant was afebrile and asymptomatic, and the infection was detected only by close visual inspection. The participant was treated with twice daily antibiotic for the 7 days prior to the device removal. Pedestal site skin cultures taken at device removal revealed pansensitive staphylococcus lugdunesis and staphylococcus capitis, and yeast, and appropriate antimicrobial treatment was provided. No organisms grew from cultures taken of adjacent bone. The only macroscopic evidence of infection at device removal was a small area (~2 cm³) of erythema and friable tissue at the skin adjacent to the right pedestal. Details of the surgical implantation and

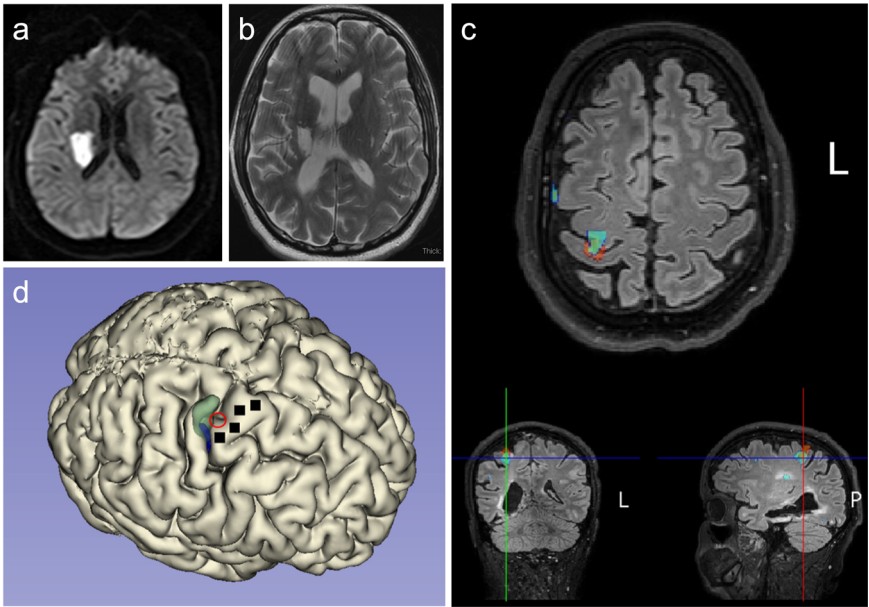

**Fig. 3 Neuroimaging results.** Diffusion sequence when the acute stroke occurred; diffusion restriction is evident in the right lentiform nucleus and adjacent white matter (**a**). T2-weighted MRI two years later shows areas of encephalomalacia and relative ventriculomegaly (**b**). Functional neuroimaging revealed a hot spot of activation, indicated by a red circle, in the depth of the central sulcus along the 'hand knob' area of the precentral gyrus (**c**). A three-dimensional reconstruction of the participant's cortical surface derived from MRI with imagined left hand movement centroid of activity indicated by the red circle (**d**). Green shading indicates an area responsive to sensory stimulation of the left hand. Black squares indicate microelectrode arrays.

removal of the device is reported separately[28]. The participant was discharged home. The participant remains in the Cortimo trial for ongoing neurosurgical follow-up and surveillance, and to track any further performance improvements in myoelectric MyoPro use with ongoing outpatient occupational therapy.

**Preoperative anatomic and functional neuroimaging.** Preoperative imaging revealed the old infarct in right lentiform nucleus and adjacent white matter including corona radiata and a portion of the posterior limb of the internal capsule, along with a large old right PCA infarct, progressed since the acute stroke imaging MRI from 2019 (Fig. 3). In addition, a small region of bandlike signal abnormality involving subcortical white matter and medial aspect of hand knob region of right precentral gyrus was identified, likely reflecting retrograde neuronal degeneration. On DTI, there was extensive loss of fractional anisotropy in the region of right corticospinal tract from old infarct. The imagined left hand motor paradigm and passive motor paradigm were diagnostic with good concordance. Subsequent to hypercapnia challenge, a BOLD signal was evident at the precentral gyrus. On the imagined left hand motor paradigm, activation was noted in the expected location along central sulcus involving lateral aspect of the hand knob region of the precentral gyrus and the adjacent portion of postcentral gyrus (Fig. 3c). On the passive left elbow motor paradigm, activation was seen along central sulcus which shows good concordance with the imagined motor task as discussed with a slightly more posterior and superior extension of activation reflecting the prominent sensory component of this passive motor paradigm. A 3D brain model was printed using the 3D FLAIR sequence to allow for 3D visualization of the surgical field for more accurate pre-operative planning (Fig. 3d).

**Neural recordings.** Well-delineated single units were recorded from 87 of the 256 channels (Fig. 4). Neural activity correlated with actual and attempted movements in both the paretic left arm in addition to the intact right arm. The discharge rate of various units appeared to correlate with specific residual actions,

including the wrist extension that gradually developed in the course of the three-month duration (Fig. 5). By taking the spike counts recorded at each channel every 200 milliseconds and running them through a leaky integrator[29], and then summing these leaky integrator outputs across all channels, we were able to visualize the cumulative cross-array firing rate activity in comparison to forearm electromyographic activity (Fig. 6, Supplementary Fig. 1). Of the 256 electrodes, in each session, we identified 40 channels that were eventually used for neural decoding. These channels were used for extraction of neural features that coded for hand and elbow flexion and extension. Two main hand open-close decoding approaches were used: (1) A discrete two-state classifier based on a 1-dimensional linear filter continuous output; (2) a training-less threshold crossing approach with a rolling baseline normalization.

**Orthosis control.** The left upper extremity score on the Action Research Arm Test (ARAT)[19] was 0 without the orthosis on measured at 1 month pre-implant and again at 3 months post-implant (Table 1). Three months-post-implant, the ARAT was 5 using the orthosis under myoelectric control, and 10 using the orthosis under direct brain-control (myoelectric control was not measured pre-implant because only a non-customized, less comfortable version of the MyoPro was available for the introductory training period).

In one component of the Jebsen-Taylor standardized test of hand function[30], the goal is to pick up and move 5 cans, one at a time, a few inches away forward on the table (normal times are 3.23 s for empty soup cans in subtest 6, and 3.30 s for full cans in subtest 7). Because the design of the hand orthosis precluded the ability to grasp a soup can (i.e., the brace only supports the thumb and next two digits), the participant performed variations on the test. It took the participant 146 s to pick up, move and release 5 pill bottles using the orthosis under myoelectric control, and 95 s to perform the identical task using the orthosis under BCI control (both performed on post-implant day 84). Another task was – with the powered orthosis donned- to hold an object in the right

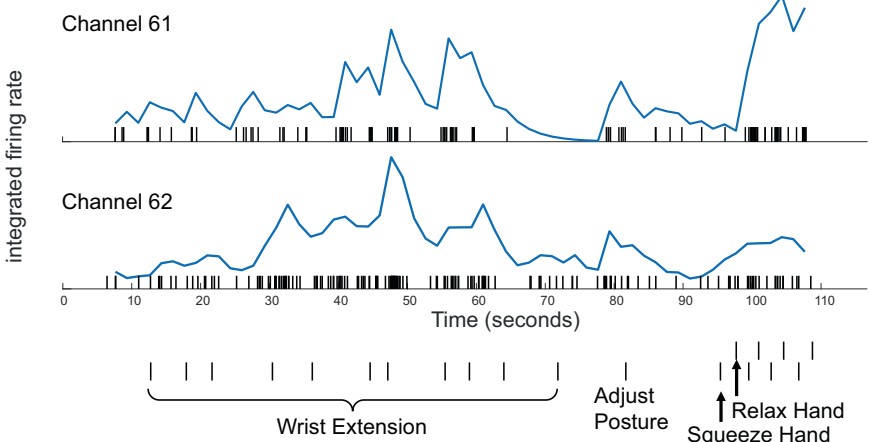

**Fig. 4 Action potential waveforms.** Snapshot of action potential waveforms recorded from two of the arrays.

**Fig. 5 Neuronal activity correlated with performed movements in the paretic limb.** Over a 110-s period, the participant was asked to perform a series of left limb movements (described on abscissa). Verbal movement instructions indicated by hash marks. Rasters indicate the time of each action potential. Normalized, integrated firing rates appear beneath each raster, derived by a 'leaky integrator' eq. in[29]; normalization achieved by dividing by the maximum integrated firing rate from each unit's spike train over the time period displayed. The top unit (channel 61) is more active for hand squeezing than wrist extension, relative to the bottom, simultaneously recorded unit (channel 62). The participant performed all movements: such motions required effort and he was unable to engage a consistent level of activity for each cue and exhibited a variable reaction time. The participant was easily fatigued, requiring him to take a break and adjust posture. Numerical data that was used to generate this figure is available in Supplementary Data 2.

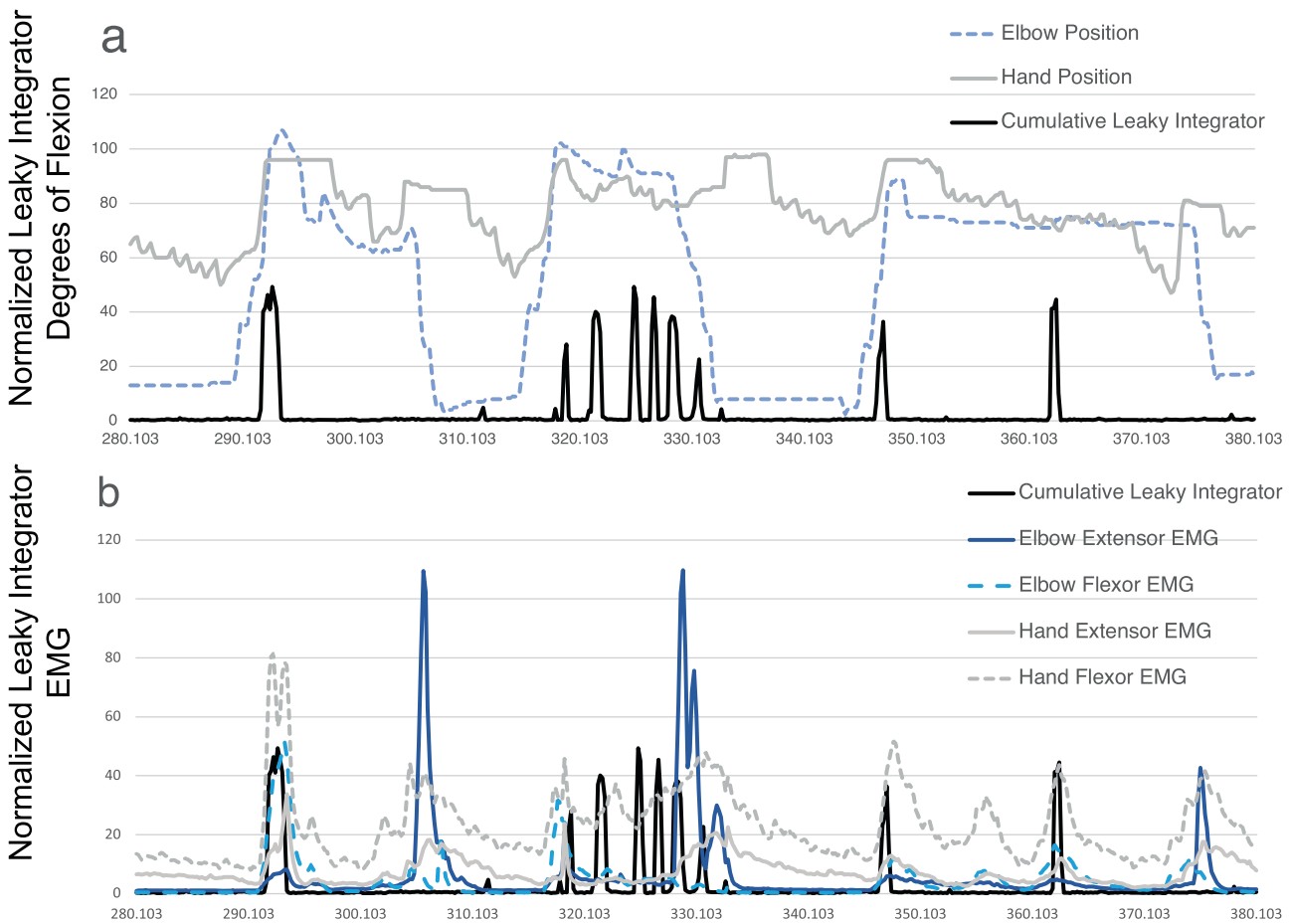

**Fig. 6 Cumulative, integrated spike activity across channels and simultaneous joint position and electromyographic activity.** The integrated neural activity fluctuated with joint position and residual left forearm electromyographic activity. The summed spike activity across channels and run through a leaky integrator[29] (solid black line) appeared to fluctuate with specific residual actions in the left upper extremity (**a**). Proximal residual activity generated a normal appearing pattern as seen between 290 and 310 s, biceps (dashed light blue line) and triceps (solid dark blue line) activity alternate (**b**). In the distal upper extremity, however, wrist flexor (dashed gray line) and wrist extensor (solid gray line) activity, tend to occur simultaneous in an abnormal manner (simultaneous agonist and antagonist contraction); also, wrist flexor activity is abnormally synergistic with biceps activity (an abnormal flexor synergy). The summed, integrated spiking activity across channels (black line) appears to covary with wrist flexor activity. Numerical data that was used to generate this figure is available in Supplementary Data 2.

hand (e.g., a stress ball or a whiteboard eraser) and place it into the paretic left hand, and then extend the left arm down towards the floor and drop the object into a bin; this process was then repeated 5 times in a row. Both these tasks were performed with the participant seated. On two trials of this pick-up-and-drop-5 with the orthosis in myoelectric mode, the participant's completion times were 222 and 128 sec/trial (performed on post-implant days 55 and 67, respectively); seconds; on five trials of the same task in BCI mode, times were 81, 106, 137, 214 s (post-implant days 48, 53, 55, 73). In addition to measuring the total time to perform these grasp-move-release tasks, we also quantified the time it took to release an object once the hand was in the target position. In addition to the five-in-a-row trials, the participant was also able to use the powered orthosis to perform numerous other single or two to four-in-a-row of the same task. Hand release times, while wearing the powered orthosis, measured over multiple trials spanning multiple days of this identical task were faster under BCI control than myoelectric control ($p = 0.04$, two-sample $t$ test; Table 2).

**Unassisted motor outcomes**. The following motor measures were performed when the participant was not connected to the BCI and was not wearing the MyoPro orthosis: Manual Muscle Testing,

Fugl-Meyer upper extremity score, Motricity Index, Stroke Impact Scale, and the Modified Ashworth Scale (Tables 3, 4, Fig. 7, Supplementary Data 1). Collectively, these serial measures demonstrated that the implantation procedure did not decrease residual strength in the paretic left arm and left leg. In fact, muscle strength increased in the left arm. Whereas serial neurological exams (performed without the assistance of any device) since the time of the stroke demonstrated an absence of voluntary wrist extension or finger extension (0/5) on manual muscle testing, starting two months into the trial, the participant began to consistently exhibit voluntary wrist extension against gravity (3/5), and on a few occasions was able to voluntarily extend the fingers slightly (2/5)[31]. The Motricity Index was 48 at one month pre-implant, 61.5 two months post-implant, and 75.5 three months post-implant. One month prior to the device implantation, the Fugl-Meyer upper extremity score was 30 (out of a maximum of 66) for the left upper extremity at baseline (one month pre-implant); this increased to a score of 36 one month after the two Multiports were implanted, and a score of 38 seven weeks post-implantation[18] (it was not possible to perform the Fugl-Meyer at exactly 8 weeks due to participant fatigue and the logistics of occupational therapist availability). The Stroke Impact Scale was 232 one month prior to implantation (the scale ranges from 64 to 320 where higher numbers indicate better

**Table 1 Arm Research Action Test.**

| CONDITION | 12 days (1.5 weeks) post-implant | 10 weeks post-implant) | 12 weeks post implant |
|---|---|---|---|
| | NO BRACE | EMG-MYOPRO | BCI-MYOPRO |
| Block 10 cm3 | 0 | 0 | 0 |
| **Block 2.5 cm3** | 0 | **1** | **1** |
| ***Block 5 cm3*** | 0 | 0 | **1** |
| Block 7.5 cm3 | 0 | 0 | 0 |
| Cricket ball | 0 | 0 | 0 |
| **Sharpening stone** | 0 | **1** | **1** |
| Pour water from one glass to another | 0 | 0 | 0 |
| **Displace 2.25-cm alloy tube from one side of table to the other** | 0 | **1** | **1** |
| **Displace 1-cm allow tube from one side of table to other** | 0 | **1** | **1** |
| **Put washer over bolt** | 0 | **1** | **1** |
| Ball bearing held between ring finger and thumb | 0 | 0 | 0 |
| ***Marble held between index finger and thumb*** | 0 | 0 | **1** |
| Ball bearing held between middle finger and thumb | 0 | 0 | 0 |
| Ball bearing held between index finger and thumb | 0 | 0 | 0 |
| Marble held between ring finger and thumb | 0 | 0 | 0 |
| Marble held between middle finger and thumb | 0 | 0 | 0 |
| Hand to behind the head | 0 | 0 | 0 |
| Hand to top of head | 0 | 0 | 0 |
| ***Hand to mouth*** | 0 | 0 | **3** |
| SUM | 0 | 5 | 10 |

The participant was unable to perform any aspect of the test with the unassisted, paralyzed arm. Conditions performed with the assistance of the MyoPro powered orthosis, whether under myoelectric or brain-computer interface control, are shown in non-italicized boldface. Conditions performed with the assistance of the MyoPro powered orthosis, only under brain-computer interface control, are shown in italicized boldface.

**Table 2 Object Release Times.**

| BCI Control | 3 4 11 2 6 18 5 17 18 13 7 26 24 2 3 2 3 1 2 5 2 9 4 2 5 9 14 7 4 12 |
|---|---|
| EMG Control | 45 8 13 24 5 1 7 19 3 4 |

Hand release times measured in seconds over multiple trials spanning multiple days of this identical task were faster under BCI control than myoelectric control ($p = 0.04$, two-sample $t$ test).

function; see Supplementary Data 1 for item-specific scores). Due to coordination error, the Stroke Impact Scale was not recorded during the implant phase; it was subsequently recorded six months post-implant (three months post-explant) as 269. Although the participant did not receive botulinum toxin injections, or receive any type of anti-spasticity medication, during the clinical trial pre-operative or implantation phase, spasticity gradually decreased with time as reflected in gradually decreasing numbers on serial measurements of the modified Ashworth scale for spasticity for passive flexion and extension movements of the fingers, wrist, and elbow, along with internal and external rotation of the shoulder (Fig. 7)[32]. In addition to the previously listed outcome measures that were performed without any device assistance, the rectified root mean square electromyographic activity was recorded by the MyoPro sensors when the MyoPro was donned and in operation and examples at various time points in the trial are shown in Supplementary Figs. 2–6; Supplementary Fig. 7 shows the time course of principle components derived from the four time series (wrist flexors, wrist extensors, elbow flexors, elbow extensors).

## Discussion
This pilot trial demonstrated that ensemble single unit activity remains active in ipsilesional cerebral cortex overlying chronic subcortical stroke. To our knowledge, this is the first report of intracortical recordings in ipsilesional cerebral cortex for a stroke above the mesencephalon. Although the corticospinal tract is also affected in conditions such as brainstem stroke and ALS, in those cases the underlying basal ganglia and other hemispheric motor control circuitry are essentially intact. This proof-of-concept study is important because it shows that an intracortical brain-computer interface approach is feasible in a class of stroke that is far more common than brainstem stroke (or ALS) and indeed is the leading cause of disability worldwide. The trial established that single neuron, movement related activity can be decoded to control a powered orthosis restoring functionally useful voluntary upper extremity movement. Importantly, this brain-computer interface system can be used simultaneously with residual intact movement, in particular in a limb with a gradient of intact to absent voluntary movement, as is common following cerebral strokes. While myoelectric approaches based upon wrist flexion did enable voluntary hand opening, this approach triggered increased muscle tone that subsequently slowed orthosis use (as the motors were opposing the abnormal tone): the BCI control mode essentially bypassed this issue and allowed motors to operate more smoothly and quickly. Electromyographic recordings demonstrated that while the participant did continue to engage wrist flexors during BCI control, the amplitude was decreased from abnormally elevated levels to more normal amplitudes.

There are many other potential ways to activate an orthosis other than using residual electromyographic activity at the wrist: more proximal muscle activity (e.g., at the shoulder), contralateral wrist activity[11], contralesional scalp EEG[13], inertial measurement units to detect minute proximal movements, proximity detectors (i.e., using RFID tags on objects to trigger the brace to open as it approaches and then close once within a target distance or using radar or other sensors mounted on the brace to detect proximity to a target object), eye gaze, and even voice activation, represent some of numerous alternative ways to peripherally activate orthoses. This trial was about proving that useful signals could be extracted from the stroke-affected hemisphere for controlling the impaired arm. We assert that there is a distinction between the arm passively following motors triggered by peripheral

**Table 3 Motricity Index, Fugl-Meyer, Stroke Impact Scale.**

| | One month pre-implant | Two months post-implant | Three months post-implant |
|---|---|---|---|
| Motricity Index (without orthosis assistance) | 48 | 61.5 | 75.5 |
| | One month pre-implant | One month post-implant | 7 weeks post-implant |
| Fugl-Meyer Upper Extremity Score (without orthosis assistance) | 30 | 36 | 38 |
| | One month pre-implant | | Six months post-implant (three months post-explant) |
| Stroke Impact Scale (without orthosis assistance) | 232 | | 269 |

**Table 4 Manual Muscle Testing.**

| Manual Muscle Testing | 15 months pre-implant | 8 months pre-implant | Day of implant (pre-op) | 7 weeks post-implant | 10 weeks post-implant | 11 weeks post-implant | 11 weeks, 3 days post-implant | Day of explant (pre-op) 3 months post-implant | 3 days post-explant |
|---|---|---|---|---|---|---|---|---|---|
| Finger flexion | | | 0 | 0 | | 3 | 3 | 4 | 3 |
| Finger extension | | | 0 | 0 | | 0 | | 0 | 0 |
| Wrist extension | 0 | 0 | 0 | 0 | 3 | 3 | 3 | 1 | 2 |
| Wrist flexion | 3 | 1 | 0 | 0 | 5 | 3 | 3 | 3 | 2 |
| Elbow extension triceps | 2 | 0 | 4 | 4 | 5 | 5 | 4 | 2 | 4 |
| Elbow flexion biceps | 1 | 1 | 5 | 5 | 5 | 5 | 3 | 4 | 3 |
| Shoulder abduction | | | 5 | 4 | 4 | | 4 | | |
| Shoulder adduction | | | 5 | | 5 | | 5 | | |

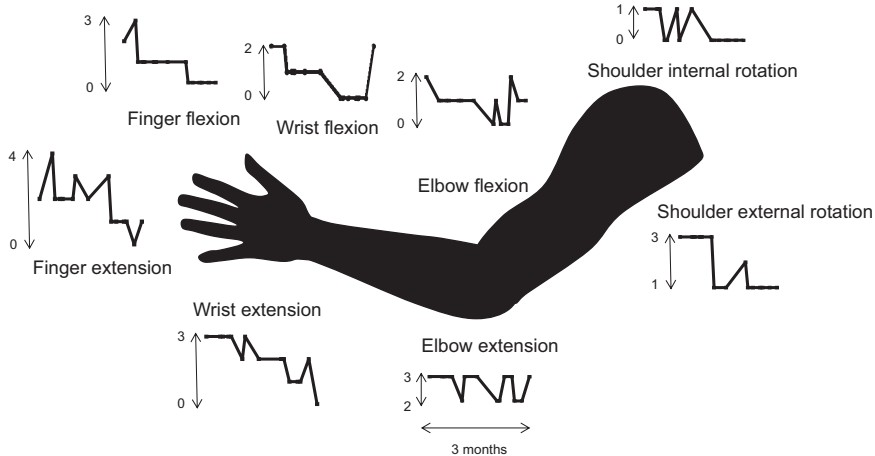

**Fig. 7 Modified Ashworth Scale variation across time.** The Modified Ashworth Scale ranges from 0 (normal) to 4, most spastic. These measurements were recorded serially over the three-month implantation phase and are shown graphically. Qualitatively, spasticity appeared to gradually decrease during this duration. Numerical data that was used to generate this figure is available in Supplementary Data 2.

sensors, versus closing the brain circuitry loop while the brain is trying to control the arm and the arm is moving. Even though of course peripheral triggers are ultimately also controlled by the brain, homing in on the actual substrate of motor control that previously had coordinated the paretic limb is presumed to exercise a plasticity that cannot be achieved by using substitute signals. The trial's demonstration justifies further exploration of motor neocortex, that has been disconnected by a subcortical stroke, as a control signal source, even if alternate peripheral modes of control exist. Indeed, it may be that a principled combination of control modes would provide patients the greatest potential for recovery.

This trial was not intended to restore voluntary motor control in the hemiparetic upper extremity in the absence of any device

use, but even so, we found that strength improved, and spasticity decreased in the native, paretic arm when BCI control was not in use. This suggests that the implantation of four arrays into ipsilesional cortex did not exacerbate pre-existing hemiparesis (i.e., it did not worsen hand or arm weakness); indeed, after the intervention hand functions improved. One potential explanation for the unexpected improvements in voluntary wrist and finger extension is mass practice. Another, more speculative, explanation for the participant's improved forearm function is that the daily exercise of ipsilesional cortical activity for BCI-orthosis control, promoted a plasticity driven response to either normalize or compensate for abnormal motor synergies.

There were several limitations in this study. It was of only one participant, only took place over a brief duration (three months),

and outcome measures could not be tested repeatedly. Most of the predetermined outcome measures (e.g., Fugl-Meyer, Motricity Index), by their nature cannot disambiguate the effects of different control modes on voluntary upper extremity use from non-specific rehabilitation effects. The one measure that could- the ARAT- was difficult in practice to perform due to the way the MyoPro hand piece clasped objects and the fact that the participant had to be tethered when the cables were plugged in. It was for this reason that the Jebsen-Taylor move-5-objects task was added.

Although the limited number of trials on various tasks reduced statistical power to compare myoelectric to BCI control, qualitatively there appeared to be a trend of faster control in the BCI mode. This may be because triggering orthosis action from direct cortical recordings does not activate abnormal forearm synergies in the same manner that myoelectric control appears to. Spasticity may represent abnormal plasticity and loss of corticoreticular facilitation of the medullary inhibition center leading to decreased inhibition from the dorsal reticulospinal tract on the spinal stretch reflex: the medial reticulospinal and vestibulospinal tracts are unopposed leading to stretch reflex hyperexcitability[33]. In the myoelectric mode, where hand closing is triggered by activation of residual wrist flexors, this hyperexcitability is inevitably triggered such that the orthosis motors must fight harder to open the hand, slowing that process. In the BCI mode, even if residual wrist flexor and extensor activity are engaged, it is to a lesser degree such that abnormal tone is not elevated, and the orthosis motors can more easily and rapidly achieve hand actions.

This pilot study implies that usable control signals are present in ipsilesional cerebral cortical activity. To be clinically scalable, future devices must be fully implantable to minimize infection risk and allow mobility. With the advent of fully implantable BCI (i.e., no percutaneous connectors[34–36]), a wider range of stroke survivors could benefit: in particular, this demonstration that usable control signals can be derived after a subcortical stroke affecting the corticospinal tract (coursing through the corona radiata) is more relevant to a wider number of people than what may be inferred in less common brainstem stroke where the supratentorial cerebral machinery of motor control typically remains intact. An option that may gain even wider clinical adoption would be to couple direct cortical control to implantable functional electrical stimulation in the paretic arm, the latter having been demonstrated in at least one person with chronic stroke[10]. Direct cortically driven peripheral muscular stimulation may have both rehabilitative[37] and direct functional benefits if deployed continuously in daily life. Fully implantable brain-computer interfaces (Fig. 8) may represent a medical device opportunity to help stroke patients break through their plateau in recovery and to achieve greater functional independence.

Previous work has demonstrated that motor cortex can continue to represent movements even years after injury has caused paralysis, such as deduced from intracortical recording in people with spinal cord injury[38], or fMRI in a person with a limb that has been amputated[39]. In hemispheric stroke, electro-myographically triggered functional electrical stimulation in the paretic limb can activate and enhance the function of the ipsilesional residual corticospinal tract[40].

We do not assert that implantable electrodes are the only or best control mode compared to alternative ways to peripherally activate orthoses: in this study, the participant served as his own control in that the orthosis was controlled by either peripheral myoelectric activity or central cortical activity. We anticipate that in many, and perhaps most, people with hemiplegic stroke that peripheral activity (e.g., inertial measurements) may suffice for orthosis control: what this current study adds is that direct central recording can also serve as a control signal source if peripheral

sources are not adequate, or indeed both types of signals could be combined. If intact substrate for motor control exists "on the other side" of a subcortical stroke, then deploying this substrate to reinstantiate control may afford an advantage over using peripheral signals that ultimately are a substitute.

The improvement that chronic stroke patients may achieve with mass practice (5 h per day, 5 days per week for 12[41,42]), raises the question of whether an invasive approach were justified. We propose that the advantage of functional restoration is that the world can be the person's rehabilitation: in other words, the person can do the rehabilitation while doing what they want to do, rather than spending hours to days every week at a rehabilitation gym. In other words, every day accumulates into mass practice and real life is rehabilitation. There are individuals with severe hemiparesis after stroke who do not regain any function even with rehabilitation (e.g., passive range of motion, stretching exercises, muscle stimulation). For this population in particular, a BCI electrical bypass could present an opportunity to regain some functional movement[43]. Such restoration would most likely be assistive in nature- namely any movement gain would cease when the BCI were not in operation. If in fact motor gains were to persist even when the BCI were turned off—implying that its use had achieved a rehabilitation effect on residual endogenous circuitry—that would suggest considerably more latent restoration potential were present after severe stroke than were currently known. If that were to be found, it might recast the use of implantable BCI as a temporary, reversible intervention to

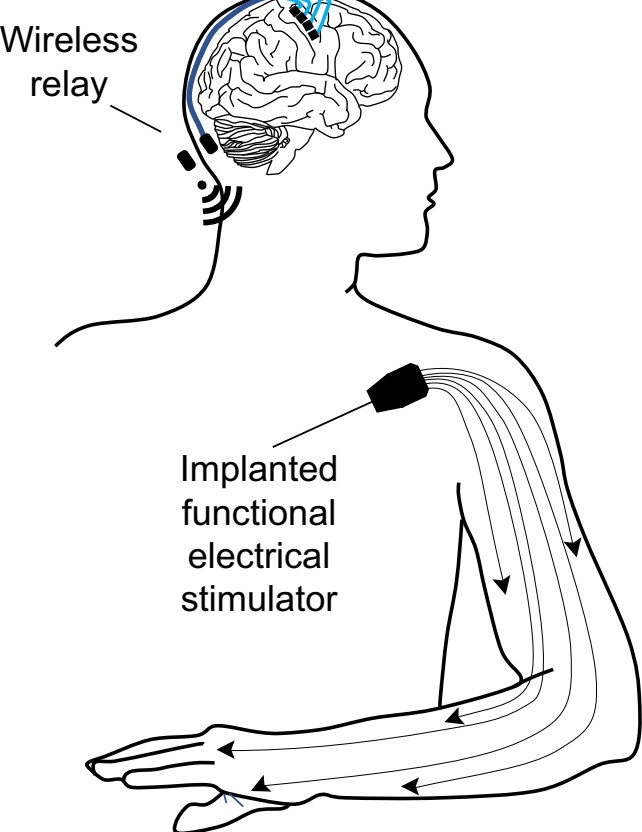

**Fig. 8 A hypothetical fully implantable upper extremity movement restoration system for people with hemiparesis from chronic stroke.** Sensors implanted within or adjacent to perilesional motor cortices relay signals wirelessly to microprocessors that decode motor intent that in turn triggers a functional electrical stimulation system implanted within the paretic limb to restore voluntary motor function.

unmask rehabilitation potential- analogous to myocardial recovery in certain with patients left ventricular assist devices that persist even after the device is removed[44].

Although wearable and implantable medical devices offer one approach to restoring motor function after stroke, stem cell[45] and surgical approaches also show promise. Surgical transfer of the C7 nerve from the nonparalyzed side to the side of the arm that was paralyzed, in adults with chronic brain injury including stroke, has been shown to yield greater improvements in function and reduction of spasticity than rehabilitation alone over a period of 12 months, when quantified by Fugl-Meyer and Modified Ashworth Scale measures[46]. In effect, this transfer approach enables a person to use the motor cortex in the intact hemisphere to achieve movement in the otherwise paralyzed limb. What that approach and the one described in this report have in common is that they both use of multi-modality therapy (e.g., passive and active exercise, physical and occupational therapy, orthoses) to leverage the person's remaining ability to learn. The approaches differ not only by the physical modality (cervical root transfer versus cortical device implant-powered orthosis), but by the areas of the brain they leverage, with the intracortical neuroprosthetic inducing a kind of exercise of the neocortex overlying the chronic stroke. In both cases, the putative mechanisms of spasticity are not directly targeted (e.g., unopposed reticulospinal and rubrospinal tone), yet both have the potential to reduce spasticity, even though in the cervical root transfer the paralyzed limb is being activated in a more natural manner than a powered orthosis pulling the limb into the desired position. Future investigation will be needed to clarify the relative benefits of each approach and the possibility of combining them together to aspire to an even greater recovery possible than one approach alone.

Ongoing progress in fully implantable multi-channel recording systems (whether subdural grids[47–49], intracortical microelectrodes[35,50] or endovascular electrodes[51,52]) to derive a control signal, and in fully implantable effector systems (whether implanted functional electrical stimulation, peripheral nerve cuffs, or epidural spinal stimulators), gives hope that the approach outlined in this report could be distilled into a modular medical device to address the leading cause of disability worldwide. Whether brain-computer interfaces to treat stroke are assistive in the sense that, like cochlear implants, their benefits accrue only when in operation, or are more like left ventricular assist devices, where continual use potentiates recovery in damaged structures in a rehabilitative manner that can outlast device operation, is a question that will require further research to address. We anticipate that there will be a distribution of patients such that for some, the device would be purely functional, and benefits would cease the moment the device were disengaged, and that for others, regular device might induce Hebbian plasticity and homeostatic mechanisms centrally and build muscle bulk and improve connective tissue peripherally in a manner that would persist even when the device were not in operation. Either outcome would represent an advance for medical science, and an opportunity for people living with stroke to not just break through the plateau of functional recovery, but to maintain those gains continually in daily life.

## Data availability

Source data for Figs. 5, 6 and 7 can be accessed in Supplementary Data 2. The Study Protocol is available at https://doi.org/10.5281/zenodo.636545[53] and the Analytic Code is available at https://doi.org/10.5281/zenodo.588501[54]. The individual participant data comprises those that underlie the results reported in this article after deidentification (text, tables, figures, appendices, neural and kinematic data). Beginning with publication and ending 5 years following article publication, the data will be made available to researchers who provide a methodologically sound proposal to achieve the aims of the approved proposal. Proposals should be directed to Mijail.Serruya@jefferson.edu. To gain access, data requestors will need to sign a data access agreement.

## Code availability

Custom code created for this study is freely available in an online repository[54]: https://doi.org/10.5281/zenodo.5885015.

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

## Acknowledgements

This research was supported by philanthropy to the Farber Institute of Neuroscience at Thomas Jefferson University. The authors would like to thank the following people for their assistance and input: Erica Jones, Shivayogi Hiremath, Christopher Thompson, Carlos Vargas-Irwin, John Donoghue, Nicholas Hatsopoulos, Nandini Murthy, David Weisman, Kristofer Feeko, M.J. Mulcahey, Lori Eckert, Joseph Tracy, Diana Tzeng, Daniel Graves, Ashly Parekh, Joely Mass, Thomas J. Kelly, IV, Stephen Valverde, Allison Weiss, Shaista Alam, Robin Dharia, Elan Miller, Lisa Bowman, Rodney Bell, Michael Sperling, and the participant and his mother.

## Author contributions

M.D.S. designed the Cortimo trial, wrote the IDE and IRB protocols, screened candidates, enrolled the participant, performed the manual muscle testing, Motricity Index, Stroke Impact Scale, Modified Ashworth Scale, created figures, recorded the videos, wrote and edited the manuscript, conducted all correspondence with the FDA and IRB, and supervised the study. A.N. designed the decoder strategy, prepared the fMRI instruction videos, wrote the software to calibrate and run the real-time decoder and map commands onto the MyoPro orthosis and completed the software verification and validation required by the FDA. A.N. and N.S. developed the software for data recording and real-time decoding and maintained backups and curation of code scripts and data files. M.D.S. and A.N. analysed the data. J.K. conducted the occupational therapy for both the screening and intervention phases and performed the Fugl-Meyer; J.M. conducted the physical therapy and aerobic conditioning during the implant phase. N.G. performed the Action Research Arm Test with the participant. K.T., D.M., and F.M. designed the structural and functional MRI protocol and performed the scan and imaging data analysis. M.D.S., A.N. and C.W. participated in the PreSub meeting with the FDA. A.S. and C.W. designed the surgical strategy and conducted pre-operative training and performed pre-operative and post-operative outpatient clinic evaluation and care of the participant. C.W. fabricated a 3D brain model to guide the surgical approach. A.S., C.W., M.K., and R.H.R. performed the implantation surgery. A.S., C.W., and R.H.R. performed regular post-operative checks on the participant during the entire implantation phase and in the immediate post-explantation phase. C.W. and M.K. performed the device removal surgery. R.H.R. provided institutional and management support including oversight of all post-operative inpatient and outpatient care. M.D.S., A.N., J.K., J.M., and N.G., helped revise the manuscript.

## Competing interests

Drs. M.D.S. and A.N. are inventors on a US provisional patent application that has been filed by Thomas Jefferson University on the methods described in this paper. All other authors report that they do not have any conflicts of interest with the research described.
