## [Peer Review File · Communications Medicine]

Reviewers' comments:

Reviewer #1 (Remarks to the Author):

This is an interesting n=1 case report of a patient with a chronic subcortical stroke who had implantable electrodes placed over M1 on the ipsilesional side with the goal that this would allow BCI control of a powered arm orthosis. The main results were that functional outcomes, in particular hand opening, were significantly better with BCI than for myoelectric control; mainly due to avoidance of unwanted wrist flexor activation in the latter case.

In addition, there were some gains in the patient in the absence of any BCI or myoelectric activation; albeit smaller. The authors are not able to determine whether these gains were related to the BCI or to the attendant training.

Thus, we do see here that the cortex has latent capacity to send commands to an orthosis even in patients with longstanding hemiparesis due to a subcortical lesion. I agree with the idea that some patients may well need permanent assistive devices - analogous to DBS in PD.

I have concerns however.

1. My main concern here is that there is not much biology to match the engineering. No EMG analysis, no kinematics. Instead we just get the ARAT and the FM. It is always disappointing to me that considerable sophistication goes into the engineering and neural recording in work like this and then behavioral characterization relies on old clinical scales.
2. There are many other potential ways to activate an orthosis other than using the wrist. The authors are going to have to make a better case for why implantable electrodes are the only way to go rather than alternative ways to peripherally activate orthoses.
3. There has to be more reference to other work showing preserved cortical representations in the absence of use. The work of Tamar Makin comes to my mind. The authors seem to think this is a surprise but it is not. Likewise, Jules Dewald and colleagues have shown residual CST activation in the setting of arm weight support; work which also reveals latent capacity that can be rapidly expressed.
4. finally, work by Nick Ward and colleagues, and Janice Daly and colleagues, have shown how well chronic patients respond to much larger doses of rehabilitation - why not do this instead of invasive approaches?

Reviewer #2 (Remarks to the Author):

The study is potentially very interesting but the authors need to provide more information about the methods and the results. Everything is too "sketchy" in the current version.

Here some examples:

1. Methods: what's about the clinical assessment of the patient at the beginning?
2. Methods how the EMG control is achieved (decoding) and then used to restore movements?
2. Results: just mentioning the ARAT "incidentally" is not enough...are there functional tests? videos?
3. Results: is there a voluntary functional improvement after the therapy?
4. Results: any additional information about cortical reorganization?

Reviewer #3 (Remarks to the Author):

Thank you for the opportunity to review this Serruya et al., Neuromotor Prosthesis to Treat Stroke-Related Paresis for consideration in Communications Medicine. In this paper, the authors present a single case report of a person with upper limb hemiparesis after chronic stroke using an intracortical microelectrode array based BCI implanted in ipsilesional precentral gyrus to control an elbow-wrist-hand orthosis that opened and closed the affected hand. The authors found that the participant could acquire voluntary control using the BCI with improvements in hand function while using the BCI as compared to without the device and as compared to myoelectric control.

Overall this is a well written study that shows that acquiring intracortical BCI control in chronic stroke above the brainstem is possible. The length of the manuscript is appropriate.

However, there are a number of limitations of the study that should be addressed.

- (1) First and foremost: this is a single case study showing essentially that intracortical BCI control in chronic subcortical stroke is possible. The authors stress that this is the first report of intracortical recordings in ipsilesional cerebral cortex for a stroke above the mesencephalon. The authors should further expand upon why this is a milestone and worthy of publication. Human intracortical BCI control has been demonstrated a number of times in patients with brainstem stroke and ALS, both diseases affected corticospinal tract neurons. How is a patient with subcortical stroke involving posterior limb of internal capsule anatomically so different (both brainstem and corona radiata strokes will affect descending white matter tracts) and thus why does simply showing that BCI control is possible in this case important?
- (2) The authors both oversell and undersell their results in the abstract and introduction. These sections setup the reader to believe that the trial will demonstrate “recovery” for the patient, i.e. rehabilitative BCI. That is, after the BCI was explanted, we were expecting to see continued gains in function (i.e. Fugl-Meyer and ARAT). The authors actually do describe some gains in the “Motor Outcomes” section of results (the Fugl-Meyer increased from 30 to 38 from pre- to post- implant) but these are not commented any further on in any section. Furthermore, the discussion does not provide adequate context to where this technology is headed? Is this intracortical BCI design in stroke for assistive or rehabilitative purposes? Stepping back, what was the hypothesis of the study- was it that this intracortical system would improve function, when not in BCI use, i.e. assistive BCI? Or that it would be simply possible/feasible to use such a system in subcortical chronic stroke?
- (3) Decoder design is not adequately described in methods. It seems like during training sessions, the authors asked the participant to do a variety of different motor imagery and activity tasks (open and closing the paretic hand, passive flexing and extending the elbow, opening and closing the hand, observing the computer cursor moving), from which the authors built a “linear filter”. What exact type of linear filter was this? A Kalman filter? The authors then map, via a vertical cursor acquisition task, the output of the BCI to the aperture of the hand on the MyoPro’s hand brace motor. Why were motor actions & imagery of so many different motor actions (elbow + hand movements) mapped, in this particular way, onto simply opening and closing the hand?
- (4) Related to above, the MyoPro brace has a number of joint controls- why did the authors not attempt to control other joints using BCI control?
- (5) The clinical outcome measures in the section “motor outcomes” are limited by heterogeneity of times collected with respect to implant. What we would really like to see here is a potentially association between number of sessions/trials on BCI control related to improvements in motor outcomes. The results are currently not presented as such.
- (6) Figure 6 and the associated text is confusing. What is the point of showing that there is a rough correlation between integrated spike activity across the MEA and gross movements of hand and

elbow flexion? Why are wrist flexor and wrist extensors called an “abnormal synergy” in the figure legend (synergies in chronic stroke have a very particular meaning and not sure what a synergy between different movements of one joint would mean)?

(7) The comparison of myoelectric control with BCI control is a nice part of the paper. I would stress this more. It shows that BCI control, in subcortical stroke, has potential benefits over peripheral nervous system control. There is some dedicated discussion to this in discussion (lines 438-450) but this is limited. The authors should consider reframing the observed improvement in BCI control over myoelectric control as the scientific focus of the paper.

(8) The presentation of the study would greatly benefit from a movie showing pre session motor function, during session motor activities, and post session motor function for this participant.

Overall, this study has potential but it currently falls in-between an engineering paper and a neuroscience paper as well as between an assistive and rehabilitative BCI study; and gets lost in this middle ground. There is no clear hypothesis presented. The decoder design and engineering aspects are not rigorously presented. The discussion and implications for why an intracortical BCI is important in stroke (and additive to other types of restorative treatments for patients with stroke) is not adequate.

Reviewer #4 (Remarks to the Author):

This is an n-of-1 safety and feasibility study to test whether a wearable, powered exoskeletal orthosis, driven by a percutaneous, implanted brain-computer interface (BCI), using the activity of neurons in the precentral gyrus in the affected cortical hemisphere, could restore voluntary upper extremity function in a person with chronic hemiparesis subsequent to a cerebral hemispheric stroke of subcortical gray and white matter and cortical gray matter. The introduction and background in this area of research was succinct and thorough laying out the pros and cons of other less-invasive systems. In contrast, the methods and results section left out some of the most critical information that if included, would improve the quality and understandability of this n-of-1 feasibility study. AE's were reported as expected for any safety n-of-1 study.

Major Concerns:

- 1) Generally well-written, readability could be improved with the inclusion of a table that summarizes the scores on each pre-specified outcome measure by time point (e.g. pre-implant, during 3 months of training, after device removal).
- 2) No mention of what was done during the OT/PT in the 6 weeks leading up to the implant. On Figure 1 it says OT screening, but there is no description of what screening was done.
- 3) Pg 6 No mention of whether or not the clinicians were trained and standardized on administration of the pre-specified outcome measures. A change in a given measure would be more credible if the clinicians were trained and standardized in administration. Otherwise a change in score could be due to inconsistency in test administration and not improvement on the test as performed by the patient.
- 4) Pg 6, Recording sessions—please clarify that the first two months (~40 sessions) were spent filter building and structured clinical end-point trials. Then the last month (~20 sessions) was used with the participant proceeding directly to BCI-controlled hand action after cables were connected. There is no mention of how long the sessions lasted. Was the participant given homework during the off-recording times? Please clarify what is meant by the statement, “The electrodes and neural signals selected immediately before filter building remained constant for any given session’s orthosis

control trials.”

5) Pg 7 during the implant study period the patient received concomitant OT and PT. How long were the sessions, what specific tasks were practiced, were the sessions logged? Were the movements timed or quantified in any way? What is meant by Restorative Therapies? XCite? Please provide a reference for vibration therapy for spasticity management.

6) Pg 8 line 284-285 suggests there is an expectation for performance improvement in MyoPro use + ongoing OT after device removal, yet this is not stated explicitly. The timeline does show routine follow-up for 9 months. What is the purpose of this 9 month follow-up?

7) Pg 9 line 355 mention of the Jebsen-Taylor standardized test of hand function...this test was not mentioned as a pre-specified outcome measure. Please clarify how it was used. As an outcome or a training tool?

8) Pg 9, For the section on Orthosis control, it would be good to know which of the 19 items on the ARAT comprised the 10 score using direct brain control and which items comprised the 5 score using myoelectric control. In this case, actual item scores would be more informative than the total score for describing what the patient could do using BCI. This would provide more resolution to the statement, “ability to acquire voluntary control”. What can they do that they could not do before the implant and 3 months of training with OT and PT?

9) Pg 10. Times to complete the pick-up and drop 5 look like fatigue set in if these times are consecutive (81 to 214s). Did fatigue subside over the 3 month implant study period. The authors state on pg 10, line 380-384 that the “Hand release times were faster under BCI control than myoelectric control, but the actual release times are not reported. (This is an example of the inconsistencies in reporting throughout the results; pick-up and drop 5 times are reported earlier in the same section but not at the end, only the t-test, p value).

10) Is the difference between 5 (myoelectric control) and 10 (BCI-control) on the ARAT meaningful? Again, providing details of which items were different in the two cases. This could also support the idea that under myoelectric control, flexor spasticity interfered with smooth operation of the motors (motors were opposing the abnormal tone). This would support the statement on pg 11, line 423.

11) Patient-specific vs generalizability to a “wider range of stroke survivors” with a fully implantable device. This may be an overstatement. The authors should mention that occlusion of the posterior cerebral artery is not as common as the more typical MCA occlusion. Preservation of usable control signals in ipsilesional cerebral cortical activity should not be assumed with typical MCA occlusion, especially given its distribution to the motor areas where the implant is placed. Therefore, a critical eligibility criterion should include the same preoperative anatomic and functional neuroimaging described here.

12) Pg 10, line 400—was this the entire SIS or just part? Prespecified outcomes were the Hand and Recovery Scales within the SIS...not sure which score goes with which—i.e. SIS score 232 one month prior to implantation, six months post implantation score 269. Providing the item scores are more useful than the total score. Then the reader can determine what specifically changed—the Hand and/or Recovery Scale.

13) Pg 10, line 405-408, What were the Ashworth Scale scores?

14) Pg 11, line 417 “restored functionally useful voluntary upper extremity movement...needs qualifier...with a BCI—orthosis.

15) Pg 11, line 433—the authors mention “mass practice” to explain the unexpected improvement in voluntary wrist and finger extension, but there is no metric of mass practice—e.g. number of repetitions, session times, time on target etc. In fact several lines later, line 438 a contradictory statement is made about “the limited number of trials on various tasks...”

16) Finally, what was the participant’s response to the acquired voluntary control over the hand-orthosis BCI?

Minor concerns:

Pg 3 line 84-85 change ...rich sources to rich source of high resolution

Pg 4 line 106, a comma is needed between 17 and 18 references above "technique".

Pg 9 line 351 delete a at the end of the lin

September 17, 2021

Dear Colleagues:

Reviewer 1:

My main concern here is that there is not much biology to match the engineering. No EMG analysis, no kinematics. Instead we just get the ARAT and the FM. It is always disappointing to me that considerable sophistication goes into the engineering and neural recording in work like this and then behavioral characterization relies on old clinical scales.

We agree that a potential benefit of a neuroengineering study such as this one, is that it can provide objective data to help characterize validated clinical scales. Although we recorded EMG (rectified RMS) activity (biceps, triceps, wrist flexors/extensors) and kinematics (elbow and wrist position) longitudinally, it is unclear which analyses would be most meaningful in the context of this report. Additional figures have been added including: 1) graphs showing concurrently recorded wrist/elbow position, the four EMG traces, example spike rasters, and the population cumulative signal; 2) example EMG recorded at different tasks on different dates; 3) EMG principal components plotted over the trial duration.

2. There are many other potential ways to activate an orthosis other than using the wrist. The authors are going to have to make a better case for why implantable electrodes are the only way to go rather than alternative ways to peripherally activate orthoses.

We have added the following text to the Discussion:

There are many other potential ways to activate an orthosis other than using residual electromyographic activity at the wrist: more proximal muscle activity (e.g., at the shoulder), contralateral wrist activity¹¹, contralesional scalp EEG¹³, inertial measurement units to detect minute proximal movements, proximity detectors (i.e., using RFID tags on objects to trigger the brace to open as it approaches and then close once within a target distance or using radar or other sensors mounted on the brace to detect proximity to a target object), eye gaze, and even voice activation, represent some of numerous alternative ways to peripherally activate orthoses. This trial was about proving that useful signals could be extracted from the stroke-affected hemisphere for controlling the impaired arm. We assert that there is a distinction between the arm passively following motors triggered by peripheral sensors, versus actually closing the brain circuitry loop while the brain is trying to control the arm and the arm is moving. Even though of course peripheral triggers are ultimately also controlled by the brain, homing in on the actual substrate of motor control that previously had coordinated the paretic limb is presumed to exercise a plasticity that cannot be achieved by using substitute signals. The trial's demonstration justifies further exploration of perilesional motor neocortex as a control signal source, even if alternate peripheral modes of control exist. Indeed, it may be that a principled combination of control modes would provide patients the greatest potential for recovery.

The participant in this trial was asked to trigger MyoPro motor actions both while the orthosis was sitting to the side, and when it was donned on his paretic arm. Although he was ultimately able to control it as

quickly when it was donned, it took him longer to learn how to achieve this.

3. There has to be more reference to other work showing preserved cortical representations in the absence of use. The work of Tamar Makin comes to my mind. The authors seem to think this is a surprise but it is not. Likewise, Jules Dewald and colleagues have shown residual CST activation in the setting of arm weight support; work which also reveals latent capacity that can be rapidly expressed.

We thank the reviewer for highlighting the work of these colleagues: discussion of their work has now been added to the Discussion:

Previous work has demonstrated that motor cortex can continue to represent movements even years after injury has caused paralysis, such as due to spinal cord injury³³, or a limb has been amputated³⁴. In hemispheric stroke, electromyographically triggered functional electrical stimulation in the paretic limb can activate and enhance the function of the ipsilesional residual corticospinal tract³⁵.

We could not find the exact text from which the reviewer concluded that we were surprised by preserved cortical representations. We in fact were not surprised: the first author worked with the first human chronically implanted with a multi-electrode array in the precentral gyrus- several years after a cervical spinal cord injury- and demonstrated that representations for a variety of movements were preserved. Although the work of the Dewald team clearly demonstrated the ability to rapidly activate the residual CST, short of having done the study we report, we do not know how else one could have proven that a person with chronic cerebral stroke could achieve the ability to drive a powered orthosis via modulation of directly recorded perilesional neuronal ensembles.

4. Finally, work by Nick Ward and colleagues, and Janice Daly and colleagues, have shown how well chronic patients respond to much larger doses of rehabilitation - why not do this instead of invasive approaches?

We thank the reviewer for highlighting the work of the Ward and Daly groups and the reviewer's question about the justification for an invasive approach. We have added the following text to the discussion:

The improvement that chronic stroke patients may achieve with mass practice (5 hours per day, 5 days per week for 12 weeks)³⁶, raises the question of whether an invasive approach were justified. We propose that the advantage of functional restoration is that the world can be the person's rehabilitation: in other words, the person can do the rehabilitation while doing what they want to do, rather than spending hours to days every week at a rehabilitation gym. In other words, every day becomes "mass practice" and "real life is rehabilitation."

Furthermore, as work by Ward and colleagues have shown, chronicity does have a mediating effect on how well patients respond to intense practice. Another outstanding question about mass practice is how durable it is, in particular for chronic patients: if it wears off quickly then even if it safer than an invasive procedure, it becomes impractical and medically futile, and we end up at the current standard of care where rehabilitation therapists help the person live with their "new normal" even if it is not as optimal as

what may be achieved by months of non-stop practice. We do not see why mass practice and invasive approaches would be mutually exclusive: for certain patients, it may be that device-based approaches paired with rehabilitation offer the greatest recovery potential (as shown by Dewald and colleagues), and invasive approaches continue to “live with the patient.” In some ways, this question about the justification of potentially fatal surgical risk is akin to that for cochlear implants: a child with sensorineural deafness is not expected to die due to being deaf and could live a healthy, meaningful life using one or more forms of sign language, whereas the surgery could potentially be life-threatening. Yet certain families feel that risk is justified. Likewise, intensive rehabilitation undoubtedly helps patients with Parkinson’s disease, and yet there is value for a deep brain stimulator in certain cases, even if it does not reverse the underlying neuropathophysiological process. We do not approach brain-computer interfaces ideologically (i.e., “invasive=better”): instead, we are trying to understand what is possible, so we have more options for our patients with chronic disabilities. If peripheral devices and mass practice can help bring lasting improvements in everyday life, we whole-heartedly applaud and encourage that. If fully-implantable brain-computer interfaces can be shown to be safe and effective, we anticipate that this could be a useful intervention for certain patients and this report is just one more step along a long journey for our field.

Reviewer 2:

1. Methods: what's about the clinical assessment of the patient at the beginning?

The following text has been added line 140:

Clinical assessment of the participant included neurological exams one year, six months and one month prior to enrollment in the setting of routine outpatient care; the exam was serially repeated once enrolled. Clinical assessment included detailed history review and confirmation of meeting all selection criteria. The trial was constructed to include a six-week screening phase (Figure 1), during which the participant underwent occupational therapy (1 hour per session, three times per week) to assess how well the participant could understand and master use of the MyoPro device. The neuropsychological testing also took place during the screening phase. Predefined outcome measures (described subsequently) were also recorded during the screening phase.

2. Methods how the EMG control is achieved (decoding) and then used to restore movements?

To address this reviewer’s comment, we have added the following text to the Methods section:

MyoPro EMG control. The MyoPro device is designed to use four EMG channels to achieve control user’s control over two electric motors that control hand and elbow motion. The four EMG channels are fixed and they are:

- 1. Forearm extensors*
- 2. Forearm flexors*
- 3. Triceps*
- 4. Biceps*

It is important noticing that the MyoPro does not use raw EMG signals for hand/elbow control mechanisms, but it calculates in real-time the rectified RMS of the raw EMG signals and the RMS signals are eventually compared against the user selected thresholds for motion classification.

Depending on the control mode selected, a combination of the above channels can be used to control either a single joint at a time or both. Namely, the available modes are: dual, open or close, more details regarding their characteristics are introduced below:

- **DUAL MODE:** Hand motion controlled by two manually set thresholds based on forearm EMGs. Elbow motion controlled by two different manually set thresholds based on bicep and tricep EMGs. Control principle is as described in the equation below:

$$\text{Motion Type} = \begin{cases} \text{Open,} & | \text{ Flexors}_t < \text{FlexTh} \quad \text{AND} \quad \text{Extensors}_t > \text{ExtTh} \\ \text{Hold,} & | \text{ (Flexors}_t < \text{FlexTh} \quad \text{OR} \quad \text{Extensors}_t < \text{ExtTh}) \\ & \text{OR} \\ & \text{(Flexors}_t > \text{FlexTh} \quad \text{OR} \quad \text{Extensors}_t > \text{ExtTh}) \\ \text{Close,} & | \text{ Flexors}_t > \text{FlexTh} \quad \text{AND} \quad \text{Extensors}_t < \text{ExtTh} \end{cases}$$

Equation 3. MyoPro DUAL Mode Motion Control Strategy. In DUAL mode the MyoPro uses flexors and extensors for each of the two joints (hand and elbow) and two thresholds for motion control decision making.

- **OPEN:** Hand motion controlled by one manually set threshold based on forearm extensors. Elbow motion controlled by one different manually set threshold based on triceps EMGs. Control principle is as described in the equation below:

$$\text{Motion Type} = \begin{cases} \text{Open,} & \text{Extensors}_t > \text{ExtTh} \\ \text{Close,} & \text{Extensors}_t < \text{ExtTh} \end{cases}$$

Equation 4. MyoPro OPEN Mode Motion Control Strategy. In OPEN mode the MyoPro moves the joint into full flexion if the extensors are below the extension threshold while the joint is moved into full extension if extensors are above the extensor threshold.

- **CLOSE:** Hand motion controlled by one manually set threshold based on forearm flexors. Elbow motion controlled by one different manually set threshold based on biceps. Control principle is as described in the equation below:

$$\text{Motion Type} = \begin{cases} \text{Close,} & \text{Flexors}_t > \text{FlexTh} \\ \text{Open,} & \text{Flexors}_t < \text{FlexTh} \end{cases}$$

Equation 5. MyoPro CLOSE Mode Motion Control Strategy. In CLOSE mode the MyoPro moves the joint into full extension if the flexors are below the flexion threshold while the joint is moved into full flexion if extensors are above the flexion threshold.

where Flexors and Extensors are the corresponding EMG signals and ExtTh and FlexTh are respectively manually set thresholds for extensors and flexors.

3. Results: just mentioning the ARAT "incidentally" is not enough...are there functional tests? videos? A table of manual muscle testing (Table 5) and a figure showing the modified Ashworth spasticity scale (Figure 7) have been added. A series of videos have been added.

Results: is there a voluntary functional improvement after the therapy?

If the reviewer means by this whether voluntary movement independent of wearing the orthosis improved, the answer appears yes: a table showing serial manual muscle testing suggests motor strength increased during the implantation phase of the trial.

If the reviewer means voluntary function while using the orthosis then this is what the ARAT and modified Jebsen-Taylor tasks, and videos, sought to show.

There were improvements on the Fugl-Meyer and Stroke Impact Scale (both measured when the orthosis was not being worn): we cannot disambiguate whether such improvements were due to ongoing regular occupational and physical therapy, or whether they were due to BCI use per se.

As stated in the Discussion, the goal of this trial was not to improve voluntary function in the absence of using the orthosis. The fact that performance- by all metrics recorded (MMT, Ashworth, Fugl-Meyer, ARAT, Stroke Impact Scale)- improved suggests to us at the very least that the implantation of four microelectrode arrays into perilesional motor cortex did not impair the participant's pre-existing residual left arm function.

Results: any additional information about cortical reorganization?

We appreciate this question and share the reviewer's curiosity. We do not have this information available at this time. We did repeat the fMRI testing after device explantation and confirmed that the same areas that showed BOLD changes pre-operatively in response to imagined left hand opening and passive left hand touch were stable. We do intend to analyze the neuronal ensemble state space trajectory over the three-month recording duration- both in and of itself and in relation to peripheral arm joint positions and EMG activity – and we anticipate that this could provide some information about cortical reorganization (or to use Tamar Makin's framework, unmasking of latent connectivity and homeostasis).

Reviewer 3:

1. First and foremost: this is a single case study showing essentially that intracortical BCI control in chronic subcortical stroke is possible. The authors stress that this is the first report of intracortical recordings in ipsilesional cerebral cortex for a stroke above the mesencephalon. The authors should further expand upon why this is a milestone and worthy of publication. Human intracortical BCI control has been

demonstrated a number of times in patients with brainstem stroke and ALS, both diseases affected corticospinal tract neurons. How is a patient with subcortical stroke involving posterior limb of internal capsule anatomically so different (both brainstem and corona radiata strokes will affect descending white matter tracts) and thus why does simply showing that BCI control is possible in this case important?

The following text has been added at line 456:

Although the corticospinal tract is also affected in conditions such as brainstem stroke and ALS, in those cases the underlying basal ganglia and other hemispheric motor control circuitry are essentially intact. This proof-of-concept study is important because it shows that an intracortical brain-computer interface approach is feasible in a class of stroke that is far more common than brainstem stroke (or ALS) and indeed is the leading cause of disability worldwide.

2. The authors both oversell and undersell their results in the abstract and introduction. These sections setup the reader to believe that the trial will demonstrate “recovery” for the patient, i.e. rehabilitative BCI. That is, after the BCI was explanted, we were expecting to see continued gains in function (i.e. fugl-meyer and ARAT). The authors actually do describe some gains in the “Motor Outcomes” section of results (the fugl-meyer increased from 30 to 38 from pre- to post- implant) but these are not commented any further on in any section. Furthermore, the discussion does not provide adequate context to where this technology is headed? Is this intracortical BCI design in stroke for assistive or rehabilitative purposes? Stepping back, what was the hypothesis of the study- was it that this intracortical system would improve function, when not in BCI use, i.e. assistive BCI? Or that it would be simply possible/feasible to use such a system in subcortical chronic stroke?

We thank the reviewer for sharing this feedback. We added to the Abstract in results the sentence: “Improvements were also seen in manual muscle testing, the modified Ashworth spasticity scale, the Stroke Impact Scale and the Fugl-Meyer scale.” and in Conclusions: “The improvements in all clinical motor scales tested implies that the implantation of multi-electrode arrays into perilesional cortex does not disrupt residual activity.” As we stated in the introduction: “ A proof-of-concept that a brain-computer interface, based on micro-electrode arrays implanted in intact cortex above a subcortical stroke, could restore behaviorally useful independent, voluntary movement, could lead to the development of a fully implantable medical device that, in principle, could reverse the motor deficits caused by stroke,” hence we did not expect that use of the system would lead to functional benefits that would outlast the device being turned off or being explanted, just as we would not expect a phrenic pacer, cochlear implant or deep brain stimulator to induce enduring benefits even after they were switched off. We thus added a sentence at line 95: “The purpose of this study was to show whether an assistive brain-computer interface, when in use, could provide a behaviorally useful benefit in motor function.” While we do anticipate that regular use of an implanted BCI system- linked to actual limb movement – could achieve a rehabilitation effect (analogous to how a left ventricular assist device can allow some recovery of the myocardium in a person with end-stage heart failure), the narrow aim of this proof-of-concept study was only to show that an assistive approach was possible: as the reviewer states, our goal was that “it would be simply possible/feasible to use such a system in subcortical chronic stroke.” As we explained in the Discussion,

line 496, our intent was not to improve motor or behavioral function in the absence of device use. We cannot disambiguate whether the benefits seen in motor scales were due to BCI use or the mass practice of physical and occupational therapy integrated into the trial. We felt the most conservative conclusion was to simply assert that the neurosurgical procedure of implanting multi-electrode arrays in perilesional cortex did not cause any new impairment. That said, we do anticipate a potential rehabilitative-beyond-assistive benefit for fully implantable BCI systems, see new text added at line 548: We propose that the advantage of functional restoration is that the world can be the person's rehabilitation: in other words, the person can do the rehabilitation while doing what they want to do, rather than spending hours to days every week at a rehabilitation gym. In other words, every day accumulates into "mass practice" and "real life is rehabilitation."

To address the reviewer's points, we also added an ultimate paragraph:

Ongoing progress in fully implantable multi-channel recording systems (whether subdural grids, intracortical microelectrodes or endovascular stent-electrodes) to derive a control signal, and in fully implantable effector systems (whether implanted functional electrical stimulation, peripheral nerve cuffs, or epidural spinal stimulators), gives hope that the approach outlined in this report could be distilled into a modular medical device to address the leading cause of disability worldwide. Whether brain-computer interfaces to treat stroke are assistive in the sense that, like cochlear implants, their benefits accrue only when operation, or like left ventricular assist devices, continual use potentiates recovery in damaged structures in a rehabilitative manner that can outlast device operation, is a question that will require further research to address. We anticipate that there will be a distribution of patients such that for some, the device would be purely functional and benefits would cease the moment the device were disengaged, and that for others, regular device use would induce Hebbian plasticity and homeostatic mechanisms centrally and build muscle bulk and improve connective tissue peripherally in a manner that would persist even when the device were not in operation. Either outcome would represent an advance for medical science, and an opportunity for people living with stroke to not just break through the plateau of functional recovery, but to maintain those gains continually in daily life.

(3) Decoder design is not adequately described in methods. It seems like during training sessions, the authors asked the participant to do a variety of different motor imagery and activity tasks (open and closing the paretic hand, passive flexing and extending the elbow, opening and closing the hand, observing the computer cursor moving), from which the authors built a "linear filter". What exact type of linear filter was this? A Kalman filter? The authors then map, via a vertical cursor acquisition task, the output of the BCI to the aperture of the hand on the MyoPro's hand brace motor. Why were motor actions & imagery of so many different motor actions (elbow + hand movements) mapped, in this particular way, onto simply opening and closing the hand?

In order to address this comment, we added the text below to the Methods section:

Decoder Design. The Cortimo system has been designed to provide a series of real-time decoding methods. Namely, two types of decoders have been implemented both discrete and continuous (i.e. filters). The available discrete classifiers were Linear Discriminant Analysis (LDA), Decision Trees, Support Vector Machine (SVM), k-nearest neighbors (KNN); while the available continuous decoders were: a position-velocity Kalman filter ²⁴ and a linear filter ²⁵. All decoders could be quickly trained using data and labels recorded during training sessions (“filter building sessions”).

The Cortimo participant and the BCI system achieved the best brace control performance when the linear filter was used and the training sessions were guided by a one or a two-dimensional cursor control task. Briefly, a cursor was displayed on the computer screen and the cursor position was controlled using a weighted linear combination of (ad-hoc) selected features derived from real-time participant’s brain activity. The neural features mostly used in this trial were either the cumulative (across all selected channels) spike count binned in 200 ms windows or the cumulative (across all selected channels) spectral density of Local Field Potentials (LFPs) calculated in the frequency range 100-500 Hz with 50 Hz frequency steps.

In order to achieve better and more reliable brace control in closed-loop tasks, the continuous (linear) filter output, originally corresponding to a specific screen location, was fed into a discretization block where the decision-making rules were either:

$$\overline{PBM}_t = \begin{cases} \text{Extension,} & x_t < AT \\ \text{Flexion,} & x_t \geq AT \end{cases}$$

Equation 1. The Cortimo Discretization Block. Discretization decision rule number 1.

where \overline{PBM}_t represents the hand motion at time step t , \overline{x}_t is the linear filter output at time step t and \overline{AT} is a position threshold chosen to maximize user’s control.

$$\overline{PBM}_t = \begin{cases} \text{Extension,} & x_t < AT_1 \\ \text{Hold,} & AT_2 < x_t \leq AT_1 \\ \text{Flexion,} & x_t \geq AT_2 \end{cases}$$

Equation 2. The Cortimo Discretizer Block. Discretization decision rule number 2.

where \overline{PBM}_t represents the hand motion at time step t , \overline{x}_t is the linear filter output at time step t and \overline{AT}_1 and \overline{AT}_2 are respectively a lower and higher position thresholds chosen to maximize user’s control.

Related to above, the MyoPro brace has a number of joint controls- why did the authors not attempt to control other joints using BCI control?

The participant exhibited some difficulties in processing speed and following complex task instructions (for example, performing sequential tasks even in the intact, dominant right arm proved challenging). In addition to these cognitive challenges, the participant had a sleep disorder introducing significant

daytime fatigue. Furthermore, although the triceps were not as strong as the biceps, their activation pattern was normal enough that myoelectric control was adequate. While we fully agree that it would have been scientifically justified to compare myoelectric versus BCI control of the elbow joint, we had very limited time and patient energy to work with, hence we focused exclusively on what he could NOT achieve well on the MyoPro to justify whether the implant could achieve any added value.

The clinical outcome measures in the section “motor outcomes” are limited by heterogeneity of times collected with respect to implant. What we would really like to see here is a potentially association between number of sessions/trials on BCI control related to improvements in motor outcomes. The results are currently not presented as such.

Neural recordings were performed daily on almost every weekday for the three-month implantation duration, totally 64 days (64 recording sessions). As described in the Methods, different decoders were tested. In addition, trying different decoders, we also evaluated different forms of feedback: cursor control, a graphical 3D avatar, triggering the MyoPro while not wearing it, and triggering the MyoPro while wearing it. The participant’s level of alertness and his ability to control the MyoPro even by myoelectric control, fluctuated significantly across days. Hence, experimental sessions were heterogeneous across most days, due to factors related to the participant and due to our trial-and-error attempt to find a decoder that was adequately intuitive enough for him to use. While the participant was engaged in some form of closed-loop neural feedback on the majority of sessions, this only translated into consistent MyoPro BCI control on a subset of sessions. The only two clinical metrics that tie directly to BCI use are the modified Jebsen-Taylor and ARAT tasks, these were recorded when the participant was actually using the BCI: the other metrics such as the Fugl-Meyer, manual motor testing etc, were done when the participant was not even wearing the MyoPro let alone using BCI. As discussed in the last section of the manuscript, we did not intend to or expect that BCI use would improve motor function when the BCI was not in use: we captured those metrics to clarify if the neurosurgical procedure would compromise residual intact function, hence the improvement we saw we feel is adequate evidence that the procedure did not in fact cause any new impairment. We do not make any claim of any type of dose-response type phenomenon for BCI use because the number of sessions in which BCI operation was effective was too small and too heterogeneous to establish such a relationship. The purpose of the study was to ask: at his best, can the participant achieve meaningful control over the orthosis with BCI control and is it equivalent or better than the type of control he can achieve myoelectrically?

Figure 6 and the associated text is confusing. What is the point of showing that there is a rough correlation between integrated spike activity across the MEA and gross movements of hand and elbow flexion? Why are wrist flexor and wrist extensors called an “abnormal synergy” in the figure legend (synergies in chronic stroke have a very particular meaning and not sure what a synergy between different movements of one joint would mean)?

The point of showing the rough correlation between integrated spike activity across the MEAs and gross hand and elbow flexion is to demonstrate that the cortical activity is biased to represent the residual and abnormally co-contracting activity. We recognize that the term ‘synergy,’ - as used by Bizzi et al and other groups- typically refers to actions across distinct joints, whether within a limb or between multiple limbs;

we have changed the term for the wrist flexor and wrist extensor as follows: “simultaneous in an abnormal manner (simultaneous agonist and antagonist contraction).” We recognize that agonist-antagonist co-contraction can be normal when trying to stabilize the limb in a fixed position and are trying to explain that co-contraction is not normal when moving the limb dynamically about a joint.

The comparison of myoelectric control with BCI control is a nice part of the paper. I would stress this more. It shows that BCI control, in subcortical stroke, has potential benefits over peripheral nervous system control. There is some dedicated discussion to this in discussion (lines 438-450) but this is limited. The authors should consider reframing the observed improvement in BCI control over myoelectric control as the scientific focus of the paper.

We thank the reviewer for this observation. We were pleased that the participant was able to achieve more fluid BCI control than myoelectric control and given the heterogeneity across sessions (in particular due to the participant’s fluctuating alertness where he would often fall asleep in the middle of a session), caused us to be more conservative in our claims. On sessions when the participant was unable to achieve consistent myoelectric control (even of the more preserved elbow activation), he was likewise unable to achieve consistent BCI control: hence we focused our comparison on data from sessions where he was able to at least achieve myoelectric control. Conservatively, we feel that we can accurately state that at his best (meaning, when he was alert and engaged), he was able to control the orthosis better in BCI mode than in myoelectric-only mode.

(8) The presentation of the study would greatly benefit from a movie showing pre session motor function, during session motor activities, and post session motor function for this participant.

We have added videos showing orthosis control under myoelectric-only versus with BCI control (“during session motor activities”). In terms of pre-session or post-session motor function, we are not exactly sure what the reviewer is requesting: a video of us performing manual muscle testing in the paretic limb? Without the orthosis on, the participant could not functionally use the left wrist or hand and could not perform any functionally useful tasks with just the residual elbow and shoulder movement. We have added some videos of the participant engaged in occupational and physical therapy sessions that took place between neural recording/BCI sessions, and strictly speaking they cannot be considered immediately before or after a recording session even if they occur on the same day as such as a session.

Reviewer 4:

1) Generally well-written, readability could be improved with the inclusion of a table that summarizes the scores on each pre-specified outcome measure by time point (e.g. pre-implant, during 3 months of training, after device removal).

Tables 2, 3, 4 and 5 have been added.

2) No mention of what was done during the OT/PT in the 6 weeks leading up to the implant. On Figure 1 it says OT screening, but there is no description of what screening was done.

Text address pre-implant therapy has been added (starting at line 145):

Clinical assessment of the participant included neurological exams one year, six months and one month prior to enrollment in the setting of routine outpatient care; the exam was serially repeated once enrolled. Clinical assessment included detailed history review and confirmation of meeting all selection criteria. The trial was constructed to include a six-week screening phase (Figure 1), during which the participant underwent occupational therapy (1 hour per session, three times per week) to assess how well the participant could understand and master use of the MyoPro device. The neuropsychological testing also took place during the screening phase. Predefined outcome measures (described subsequently) were also recorded during the screening phase. During this phase, therapy consisted of evaluation of active and passive range of motion, strength, hypertonicity, and goal setting with ADLs and IADLs. Treatment of left upper extremity spasticity was completed utilizing stretching, functional electrical stimulation, and creating a splinting schedule. This pre-implant occupational therapy was considered “screening” in that it was decided prior to enrolling the participant that proceeding to implant would only occur if the occupational therapist felt that the participant could adequately comply with the therapy. Therapeutic exercise and activity were incorporated to improve postural control and non-volitional movements with left upper extremity. Introductory use of MyoPro device was incorporated within treatment. Pre-implant physical therapy included baseline functional balance measures with interventions focused on open/closed chain strengthening, static/anticipatory/dynamic postural control and gait training.

3) Pg 6 No mention of whether or not the clinicians were trained and standardized on administration of the pre-specified outcome measures. A change in a given measure would be more credible if the clinicians were trained and standardized in administration. Otherwise a change in score could be due to inconsistency in test administration and not improvement on the test as performed by the patient.

We agree with the reviewer’s point and have clarified by adding the following text at line 227:

The outcome measures were performed by clinicians who were trained and standardized in administration, and each measure was assigned to specific co-investigators to perform serially to minimize inter-rater variability across time.

5) Pg 7 during the implant study period the patient received concomitant OT and PT. How long were the sessions, what specific tasks were practiced, were the sessions logged? Were the movements timed or quantified in any way? What is meant by Restorative Therapies? XCite? Please provide a reference for vibration therapy for spasticity management.

Starting at line 283, additional text describing therapy during the implant study period has been added:

Following device implantation, the participant continued occupational therapy, twice per week, and physical therapy, once per week, each session lasting approximately one hour. In the three-month implantation phase, the participant hence received 24 one-hour

sessions of occupational therapy and 12 one-hour sessions of physical therapy. In addition, clinical trial assistants practiced therapy exercises with the participant and accompanied him to a gym for aerobic conditioning (either stationary bicycling or NuStep combined arm and foot cycle, for 20 minutes to a target heart rate of 120 beats per minutes): these sessions were approximately one hour and were practiced on a daily basis, including weekends for a total of 91 days. Occupational therapy focused on postural training while seated and walking, donning and doffing the MyoPro, repetitive trials of hand open/close elbow flex/extend with the MyoPro, and using the MyoPro for functional activities. Timed functional electrical stimulation²⁴ (e.g., pincer grasp programs using the XCite brand FES unit from Restorative Therapies) and vibration therapy (5 to 10 minutes of focal muscle vibration) were used for spasticity management²⁵. Physical therapy exercises included scapular mobilization, progressive range of motion, weight bearing, forced use with game-related activities to encourage left UE volitional control, and aerobic endurance exercise. The exact exercises performed (passive and active range of motion stretching, neuromuscular education, electrical stimulation, orthosis use), blood pressure, and subjective pain reports, were logged for every rehabilitation session; in addition, clips of several sessions were recorded by video (see Occupational Therapy log and therapy videos in supplementary materials).

The occupational therapy log (for sessions both before and during the implant phase) and video examples of therapy session clips, are included as supplementary materials.

6) Pg 8 line 284-285 suggests there is an expectation for performance improvement in MyoPro use + ongoing OT after device removal, yet this is not stated explicitly. The timeline does show routine follow-up for 9 months. What is the purpose of this 9 month follow-up?

*In designing the trial, there was **not** an expectation for indefinite performance improvement in MyoPro use with ongoing occupational therapy. Instead, the rationale for follow-up was to see if any gains that had been accrued by the time of device removal (e.g., the cumulative effect of pre-implant and during-implant MyoPro use) would be sustained. Such follow-up would provide an indirect measure of the participant's neurological integrity and compliance with the therapy regimen, granting these two components cannot always be easily separated (i.e., decreases in MyoPro performance may reflect lack of daily practice rather than any new neurological insult or disease process progression). All that said, 9-month-follow-up, if the participant were able to sustain adequate regular practice, would allow us to answer the question of whether he could eventually meet and perhaps exceed the performance level acquired under his best BCI control with purely myoelectric control (on the ARAT and Jebsen-Taylor tasks).*

7) Pg 9 line 355 mention of the Jebsen-Taylor standardized test of hand function...this test was not mentioned as a pre-specified outcome measure. Please clarify how it was used. As an outcome or a training tool?

We thank the reviewer for this comment. The Jebsen-Taylor test was not a pre-specified outcome measure: instead, it was added during the implant phase because it became apparent that the ARAT was too time-consuming and difficult (hence discouraging for the participant even if punctuated by successes on components of the test), to repeat hence encouraging us to consider other metrics. Box and Blocks was briefly attempted, and it quickly became evident that despite trying different hand-grasp pieces on the MyoPro, it was impossible to use for that task (i.e., not only was it impossible for the participant to use the MyoPro to grasp and move the blocks in myoelectric mode, it was also found to be impossible even for able-bodied investigators to use the device in myoelectric mode because of the way the finger pieces closed). The Jebsen-Taylor outcome measure is in fact a battery of numerous activities, in particular motions that are repeated. Hence the way we used it was to find one of its components- moving five objects one at a time, one after another- and have the participant perform that Jebsen-Taylor component task numerous times, either in myoelectric-only or with BCI control of the MyoPro. Hence at line 227 we have added the following text:

During the implant phase, a component of the Jebsen-Taylor measure (picking up, moving and putting down five objects, one at a time, one after the other)²³ was added because it was found that the participant was able to perform this task more consistently and easily with the orthosis than the ARAT, inspiring greater participant motivation and engagement and hence facilitating comparison of myoelectric vs BCI control modes.

8) Pg 9, For the section on Orthosis control, it would be good to know which of the 19 items on the ARAT comprised the 10 score using direct brain control and which items comprised the 5 score using myoelectric control. In this case, actual item scores would be more informative than the total score for describing what the patient could do using BCI. This would provide more resolution to the statement, “ability to acquire voluntary control”. What can they do that they could not do before the implant and 3 months of training with OT and PT?

We thank the reviewer for this excellent point: we agree. We have added Table 2 showing the actual item scores for no-orthosis, myoelectric-controlled orthosis, and with BCI-controlled orthosis.

9) Pg 10. Times to complete the pick-up and drop 5 look like fatigue set in if these times are consecutive (81 to 214s). Did fatigue subside over the 3 month implant study period. The authors state on pg 10, line 380-384 that the “Hand release times were faster under BCI control than myoelectric control, but the actual release times are not reported. (This is an example of the inconsistencies in reporting throughout the results; pick-up and drop 5 times are reported earlier in the same section but not at the end, only the t-test, p value).

We appreciate the reviewer’s valid critical concern and question. The pick-up-and-drop trials were not consecutive within a day, and were across days: this information has now been added. We did not observe a trend of fatigue increasing or subsiding in any consistent manner over the 3-month implant period: instead, fatigue appeared to fluctuate rapidly within a day apparently due to changes in attention, motivation, prior night’s sleep, and changes in spasticity in the upper extremity. The latter was sometimes affected by posture (hence the participant was repeatedly prompted to sit upright), and often would fluctuate for no observable reason. Regarding the release-times, they are described in Table 1:

BCI Control	3 4 1 1 2 6 1 8 5 1 7 18 13 7 26 24 2 3 2 3 1 2 5 2 9 4 2 5 9 14 7 4 12
EMG Control	45 8 13 24 5 1 7 19 3 4

10) Is the difference between 5 (myoelectric control) and 10 (BCI-control) on the ARAT meaningful? Again, providing details of which items were different in the two cases. This could also support the idea that under myoelectric control, flexor spasticity interfered with smooth operation of the motors (motors were opposing the abnormal tone).

A table showing the exact items is now displayed: Table 2. It is difficult to ascertain if the difference between the two modes is meaningful: with just two measurements, a statistical comparison would not be meaningful, and likewise that limits interpretation of clinical significance. If we discount the improvement in the hand-to-mouth task as proximal shoulder activation not obviously related to BCI use, then the difference in ARAT comes down to two items: the ability to pick up and move a 5 cubic centimeter block and, the ability to pick and move a marble from one position to another. It is possible (and likely) that the reason the participant could not achieve those two items under EMG control was that flexor tone opposed the orthosis motors.

11) Patient-specific vs generalizability to a “wider range of stroke survivors” with a fully implantable device. This may be an overstatement. The authors should mention that occlusion of the posterior cerebral artery is not as common as the more typical MCA occlusion. Preservation of usable control signals in ipsilesional cerebral cortical activity should not be assumed with typical MCA occlusion, especially given its distribution to the motor areas where the implant is placed. Therefore, a critical eligibility criterion should include the same preoperative anatomic and functional neuroimaging described here.

This participant experienced more than one stroke. The etiology of paresis in the participant was due to ischemia in the right lentiform nucleus and adjacent corona radiata, and part of the posterior limb of the right internal capsule. The anterior choroidal artery provides blood supply to the posterior and ventral aspects of the posterior limb of the internal capsule, and middle cerebral artery supplies the intermediate part of the posterior limb of the internal capsule. Most of the lentiform nucleus is perfused by MCA perforators (with smaller parts of the putamen by be supplied by ACA, AChA perforators, and part of the lateral globus pallidus from Heubner’s artery and ACA, and medial globus pallidus from ICA, AChA perforators). The PCA territory occlusion was a separate, parallel process (e.g., putative shower of emboli). Complete occlusion of the MCA would indeed obliterate motor and premotor cortices on the precentral gyrus and this particular n-of-1 study cannot address whether usable control signals could be derived from remaining intact cortex more anterior or posterior to such an insult. We do assert that the anatomical location of this participant’s stroke is in fact more common: a subcortical stroke affecting primarily the basal ganglia and adjacent white matter, including fibers of the corticospinal tract (i.e., coursing through the corona radiata, as was affected in our participant’s stroke). By “wider range of stroke survivors,” we mean wider than what could be inferred from BCI in patients with brainstem stroke in which there is more localized (and devastating) disruption to the corticospinal tract that typically leaves hemispheric structures (motor and premotor cortices, basal ganglia, ventrolateral thalamus, etc) intact. To clarify we have added the text:

a wider range of stroke survivors could benefit: in particular, this demonstration that usable control signals can be derived in a subcortical stroke affecting the corticospinal tract (coursing through the corona radiata) is more relevant to a wider number of people than what may be inferred in less common brainstem stroke where the supratentorial cerebral machinery of motor control typically remains intact.

Regardless of which major cerebral artery were occluded, subcortical strokes make up the majority of strokes and hence lead to the most disability. In terms of critical eligibility criterion, we were not seeking assert that such an approach is justified yet for any patient outside the setting of a clinical trial. The purpose of this study was to demonstrate what was possible and lay the groundwork for further studies that can ascertain in greater detail what the best preoperative anatomic and functional neuroimaging ought to be to identify the individuals most likely to benefit.

12) Pg 10, line 400—was this the entire SIS or just part? Prespecified outcomes were the Hand and Recovery Scales within the SIS...not sure which score goes with which—i.e. SIS score 232 one month prior to implantation, six months post implantation score 269. Providing the item scores are more useful than the total score. Then the reader can determine what specifically changed—the Hand and/or Recovery Scale.

This was the entire SIS. A comparison table (Table 4) has been added.

13) Pg 10, line 405-408, What were the Ashworth Scale scores?

A figure has been added that graphically represents the Ashworth scores over time (it was felt that a graphic would be more informative than a table of numbers).

14) Pg 11, line 417 “restored functionally useful voluntary upper extremity movement...needs qualifier...with a BCI—orthosis.

The original sentence read: “The trial established that single neuron, movement related activity can be decoded used to control a powered orthosis that restored functionally useful voluntary upper extremity movement,” and has been rewritten: “The trial established that single neuron, movement related activity can be decoded to control a powered orthosis restoring functionally useful voluntary upper extremity movement.”

15) Pg 11, line 433—the authors mention “mass practice” to explain the unexpected improvement in voluntary wrist and finger extension, but there is no metric of mass practice—e.g. number of repetitions, session times, time on target etc. In fact several lines later, line 438 a contradictory statement is made about “the limited number of trials on various tasks...”

These details have been added (requested by other reviewers; please see above).

16) Finally, what was the participant’s response to the acquired voluntary control over the hand-orthosis BCI?

Our participant is a man of few words (by personality, not due to aphasia). When asked how he felt about BCI control his reply was: "It's OK." By his expressions, we infer that he was pleased when the orthosis responded quickly and efficiently so that he could perform actions.

Pg 3 line 84-85 changerich sources to rich source of high resolution

Corrected.

Pg 4 line 106, a comma is needed between 17 and 18 references above "technique".

Corrected.

Pg 9 line 351 delete a at the end of the lin

Corrected.

Respectfully,

Mijail Serruya, M.D., Ph.D.
Assistant Professor of Neurology
Thomas Jefferson University

Reviewers' comments:

Reviewer #1 (Remarks to the Author):

I think the authors have made a commendable effort to address reviewer concerns.

Reviewer #2 (Remarks to the Author):

Thanks for addressing my questions. The paper is fine to me.

Reviewer #3 (Remarks to the Author):

The authors have revised the paper based on reviewer comments.

However, I continue to have fundamental differences in opinion with the way that the results are presented and the conclusions.

For the results, I am not clear as to the purposes of the main figures (Figure 5 and Figure 6). Figure 5 is focused on action potentials in 2 channels, integrated firing rates, and timing of wrist extension and hand squeezing. Figure 6 shows spike activity related to joint position and EMG activity. Both figures show broad temporal correlations between neural activity and timing of movements or joint position/EMG. The purpose of both of these figures, I think, is to show that neural activity from primary motor cortex can still be decoded in stroke. But the primary purpose of this paper was to show that microelectrode arrays implanted in primary motor cortex could restore voluntary upper extremity function. With this primary purpose in mind, there should be a figure of performance metrics. For example, if the main outcome measure was the Action Research Arm Test (ARAT), there should be a figure showing multiple trials of the ARAT and what the score was on each. Or the authors could show multiple subtests of the ARAT and the score of each of these. My main concern here is that there is no main figure to justify the main aim / result of this paper.

Some of the claims made in the paper continue to be conceptually wrong. The patient had a stroke in the R basal ganglia/corona radiata and Right occipital lobe. The authors claim that the multielectrode array was implanted in "perilesional cortex" but it was actually implanted in the R primary motor cortex, arm/hand area. I'm not sure how this would qualify as "perilesional cortex".

The authors claim that they demonstrate "for the first time in a human being that ensembles of individual neurons in the cortex overlying a chronic stroke remain active and engaged in motor representation and planning". This is simply not true. For example, in Hochberg et al. 2012, two people with no functional arm control due to chronic stroke (brainstem) used neuronal ensembles activity generated by intended arm and hand movements to make point-to-point reaches and grasps with a robotic arm.

Reviewer #4 (Remarks to the Author):

The purpose of this revised single-case study, "Neuromotor Prosthetic to Treat Stroke-Related Paresis" was to demonstrate feasibility that a wearable, powered exoskeletal orthosis, driven by a percutaneous, implanted brain-computer interface (BCI), using the activity of neurons in the precentral gyrus in the affected hemisphere, could restore voluntary upper extremity function in a person with chronic hemiparesis subsequent to a cerebral hemispheric stroke of subcortical gray and white matter and cortical gray matter. The authors have sufficiently addressed my major concerns by clarifying details of the 1) therapy received pre implant and following device implantation; 2) details pertaining to the decoder design, and 3) providing tables or figures of repeated outcome measures such as the Ashworth scale, ARAT, Object release times, Motricity Index, Fugl-Meyer, SIS and the details about which items on the SIS changed, and Manual muscle testing. These additional details gave credibility to statements pertaining to "improvements in all clinical motor scales tested". I thought the author's response (in the discussion) to the query concerning the risk of such an invasive procedure vs peripherally-driven prostheses was thoughtful and important for moving the field forward. In particular, "The trial's demonstration justifies further exploration of perilesional motor neocortex as a control signal source, even if alternate peripheral modes of control exist. Indeed, it may be that a principled combination of control modes would provide patients the greatest potential for recovery." The other important point, which was lost in the previous version, but now clearly emerges is that the implantation of four arrays into ipsilesional cortex did not exacerbate pre-existing hemiparesis; "indeed, after the intervention hand functions improved." This is an important point, though the exact cause of the improvement is not clear, the authors acknowledge this as well.

Altogether, the revised manuscript with the clarified text, inclusion of tables and figures and the supplemental videos not only provides a convincing argument that such a neuromotor prosthetic is feasible to treat stroke-related paresis, but this opens up the field for further exploration by others, especially given the level of details provided that allows replication and development.

This paper underscores the importance of single-case longitudinal studies, especially in frontier areas such as BCI. In its current form, I believe this paper will have a significant impact on the field.

Carolee Winstein, PhD, PT

November 18, 2021

Dear Colleagues:

Reviewer #1 (Remarks to the Author):

I think the authors have made a commendable effort to address reviewer concerns.

We thank Reviewer #1 for the feedback.

Reviewer #2 (Remarks to the Author):

Thanks for addressing my questions. The paper is fine to me.

We thank Reviewer #2 for the feedback.

Reviewer #3 (Remarks to the Author):

The authors have revised the paper based on reviewer comments.

However, I continue to have fundamental differences in opinion with the way that the results are presented and the conclusions.

For the results, I am not clear as to the purposes of the main figures (Figure 5 and Figure 6). Figure 5 is focused on action potentials in 2 channels, integrated firing rates, and timing of wrist extension and hand squeezing. Figure 6 shows spike activity related to joint position and EMG activity. Both figures show broad temporal correlations between neural activity and timing of movements or joint position/EMG. The purpose of both of these figures, I think, is to show that neural activity from primary motor cortex can still be decoded in stroke. But the primary purpose of this paper was to show that microelectrode arrays implanted in primary motor cortex could restore voluntary upper extremity function. With this primary purpose in mind, there should be a figure of performance metrics. For example, if the main outcome measure was the Action Research Arm Test (ARAT), there should be a figure showing multiple trials of the ARAT and what the score was on each. Or the authors could show multiple subtests of the ARAT and the score of each of these. My main concern here is that there is no main figure to justify the main aim / result of this paper.

Reviewer #3 is 100% correct. We agree that multiple trials of the ARAT would be ideal, however in actuality doing the test was extremely difficult and we do not have such data: we have reported all the ARAT data that we have (Table 1). As stated in prior responses to the reviewers, this trial was challenging due to the brief duration of three months and the fact that the participant had fluctuating arousal and flagging endurance for lengthy decoding sessions. One of the unexpected challenges of the trial was how difficult it was to use the MyoPro hand piece for the ARAT and numerous other tasks we selected - under any type of control mode. For an n-of-1 study we made a decision to go with a variety of primary outcome measure (Fugl-Meyer UE, ARAT, Motricity Index, Stroke Impact Scale)

that would be measured at just three time points (baseline, towards end of 3 month BCI phase, and at study close), rather than using only one of those predetermined measure repeatedly. We spent the majority of our time determining the best way to decode the signals and find a behavioral paradigm the participant could understand and work with. This was not a “plug and play” trial where we could simply repeat tasks. As we previously stated, the participant was not consistent in energy and accuracy even using his able-bodied right hand. We did appreciate the need for repeatable tests and that is the exact reason why we incorporated the Jebsen-Taylor “move 5 pill bottles” task and added improvised task (pick up an eraser and then drop it): namely tasks that the participant was more consistently able to do with the MyoPro orthosis and with the constraints placed by having the head tethered to a one or two connecting cables. We assert that Tables 1, 2 and 3 – taken collectively- are equivalent to the “main figure to justify the main aim / result of this paper.” Although these could have presented as a graphical figure, based upon feedback from other reviewers we felt it most parsimonious to report them as tables. Table 2 reports hand release times measured over multiple trials spanning multiple days of an identical task under BCI control and under myoelectric control: while we of course would have wanted to perform this task and many others with a higher number of trials, this was the most we could achieve under the constraints of the duration of the trial, participant factors, and the challenge of identifying and deploying a decoding approach that the participant could use more easily (see section “Training-less” Mapping.)

To acknowledge Reviewer #3’s correct concerns we have added the following text to the Discussion:

There were several limitations in this study. It was of only one participant, only took place over a brief duration (three months), and outcome measures could not be tested repeatedly. Most of the predetermined outcome measures (e.g., Fugl-Meyer, Motricity Index), by their nature cannot disambiguate the effects of different control modes on voluntary upper extremity use from non-specific rehabilitation effect. The one measure that could- the ARAT- was difficult in practice to perform due to the way the MyoPro hand piece clasped objects and the fact that the participant had to be tethered when the cables were plugged in. It was for this reason that the Jebsen-Taylor “move 5 objects” task was added.

Some of the claims made in the paper continue to be conceptually wrong. The patient had a stroke in the R basal ganglia/corona radiata and Right occipital lobe. The authors claim that the multielectrode array was implanted in “perilesional cortex” but it was actually implanted in the R primary motor cortex, arm/hand area. I’m not sure how this would qualify as “perilesional cortex”.

*To address this reviewer’s concern, we changed the sentences:
in the abstract:*

“The improvements in all clinical motor scales tested implies that the implantation of multi-electrode arrays into perilesional cortex does not disrupt residual activity.”

to

*“The improvements in all clinical motor scales tested implies that the implantation of multi-electrode arrays into **motor cortex overlying subcortical stroke** does not disrupt residual activity.”*

in the Discussion:

“The trial’s demonstration justifies further exploration of perilesional motor neocortex as a control signal source, even if alternate peripheral modes of control exist,” to

"The trial's demonstration justifies further exploration of motor neocortex, that has been disconnected by subcortical stroke, as a control signal source, even if alternate peripheral modes of control exist,"

The authors claim that they demonstrate "for the first time in a human being that ensembles of individual neurons in the cortex overlying a chronic stroke remain active and engaged in motor representation and planning". This is simply not true. For example, in Hochberg et al. 2012, two people with no functional arm control due to chronic stroke (brainstem) used neuronal ensembles activity generated by intended arm and hand movements to make point-to-point reaches and grasps with a robotic arm.

We will revise "chronic stroke" to be "supratentorial, subcortical stroke" and we believe this is accurate. This is a significant difference and refers to the vast majority of people with stroke. Subcortical stroke (within the cerebrum) is the most common type of stroke. Brainstem stroke leading to a locked-in syndrome is- thankfully - extremely rare and does not comprise the bulk of disability-causing stroke. That is one of the fundamental goals of this entire project: to decide if implantable BCI is feasible in a form of a more common form of stroke that contributes to it being the number one form of disability on Earth. Locked-in-syndrome brainstem stroke, while devastating, does not impact the same number of human beings. Furthermore, decoding ensemble activity is arguably more challenging in a person with a mixture of paralysis, paresis and intact function and fluctuating spasticity, and this is the situation for the majority of human beings impaired by chronic stroke. The lead author helped pioneer the initial BrainGate trial that led to the Hochberg 2006 paper upon which the 2012 paper was based and can state unequivocally that decoding ensemble activity in a person who does have residual functional arm control is in fact more challenging than when there is no functional arm control at all.

In recognition of the limitations of our study, we have also reworded the last sentence from: "...lays the foundation for a fully implanted movement restoration system, and dramatically expands the potential utility of fully implantable brain-computer interfaces to a clinical population that numbers in the tens of millions worldwide."

to

"...lays the foundation for a fully implanted movement restoration system, and **may** expand the potential utility of fully implantable brain-computer interfaces to a clinical population that numbers in the tens of millions worldwide."

Reviewer #4 (Remarks to the Author):

The purpose of this revised single-case study, "Neuromotor Prosthetic to Treat Stroke-Related Paresis" was to demonstrate feasibility that a wearable, powered exoskeletal orthosis, driven by a percutaneous, implanted brain-computer interface (BCI), using the activity of neurons in the precentral gyrus in the affected hemisphere, could restore voluntary upper extremity function in a person with chronic hemiparesis subsequent to a cerebral hemispheric stroke of subcortical gray and white matter and cortical gray matter. The authors have sufficiently addressed my major concerns by clarifying details of the 1) therapy received pre implant and following device implantation; 2) details pertaining to the decoder design, and 3) providing tables or figures of repeated outcome measures such as the Ashworth scale, ARAT, Object release times, Motricity Index, Fugl-Meyer, SIS and the details about which items on the SIS changed, and Manual muscle testing. These additional details gave credibility to statements pertaining to "improvements in all clinical motor scales tested". I thought the

author's response (in the discussion) to the query concerning the risk of such an invasive procedure vs peripherally-driven prostheses was thoughtful and important for moving the field forward. In particular, "The trial's demonstration justifies further exploration of perilesional motor neocortex as a control signal source, even if alternate peripheral modes of control exist. Indeed, it may be that a principled combination of control modes would provide patients the greatest potential for recovery." The other important point, which was lost in the previous version, but now clearly emerges is that the implantation of four arrays into ipsilesional cortex did not exacerbate pre-existing hemiparesis; "indeed, after the intervention hand functions improved." This is an important point, though the exact cause of the improvement is not clear, the authors acknowledge this as well.

We agree with the Reviewer's observation that the exact cause of improvement is not clear, and we hope that future investigations can shed light on this question.

Altogether, the revised manuscript with the clarified text, inclusion of tables and figures and the supplemental videos not only provides a convincing argument that such a neuromotor prosthetic is feasible to treat stroke-related paresis, but this opens up the field for further exploration by others, especially given the level of details provided that allows replication and development.

This paper underscores the importance of single-case longitudinal studies, especially in frontier areas such as BCI. In its current form, I believe this paper will have a significant impact on the field.

We thank Reviewer #4 for the feedback and validation.

Respectfully,

Mijail Serruya, Ph.D., M.D.

Assistant Professor of Neurology

Thomas Jefferson University

REVIEWERS' COMMENTS:

Reviewer #3 (Remarks to the Author):

This version of the manuscript is much improved. The advances and limitations of this single case report are discussed in a much more balanced manner. I do have a few suggestions for further improvement.

- (1) The difference between assistive and rehabilitative BCI is much clearer in this version but assessments/outcome measures that were performed WITH BCI control versus WITHOUT should be made clearer THROUGHOUT. For example, the discussion "This trial was not intended to restore voluntary motor control in the hemiparetic upper extremity in the absence of any device use, but even so, we found that strength improved, and spasticity decreased". I would suggest adding to the end of this sentence either (a) "... in the native arm when BCI control was not in use" or "in the native arm after BCI explant". Throughout the Motor outcome measures paragraph of the results section, the timing with reference to post-implant is confusing. Please make reference to implant when appropriate but explant as well. Overall, it should be clear throughout which assessments were performed when and which were done with BCI control, with BCI in place but without control, and with BCI removed (either because it hadn't been implanted yet or because it was explanted).
- (2) Zheng "Trial of Contralateral Seventh Cervical Nerve Transfer for Spastic Arm Paralysis" should be cited and parallels drawn to given that this is essentially a lesion bypass approach.
- (3) The paragraph "The improvement that chronic stroke patients may achieve with mass practice..." is a good discussion but this should be expanded. There are patients with severe hemiparesis after stroke who do not regain any function even with rehabilitation(which would be passive range of motion and stretching exercises). For this population in particular, lesion bypass BCI offers an opportunity to regain some native movement and could potentially offer a rehabilitative approach. The authors should discuss this as a future direction, making sure to keep the distinction between assistive and rehabilitative BCI clear and the overlap here.

January 21, 2022

Dear Colleagues:

Reviewer #3 (Remarks to the Author):

This version of the manuscript is much improved. The advances and limitations of this single case report are discussed in a much more balanced manner.

We appreciate the input of Reviewer #3 and the acknowledgement that our changes have addressed concerns previously outlined.

I do have a few suggestions for further improvement.

(1) The difference between assistive and rehabilitative BCI is much clearer in this version but assessments/outcome measures that were performed WITH BCI control versus WITHOUT should be made clearer THROUGHOUT. For example, the discussion "This trial was not intended to restore voluntary motor control in the hemiparetic upper extremity in the absence of any device use, but even so, we found that strength improved, and spasticity decreased". I would suggest adding to the end of this sentence either (a) "... in the native arm when BCI control was not in use" or "in the native arm after BCI explant". Throughout the Motor outcome measures paragraph of the results section, the timing with reference to post-implant is confusing. Please make reference to implant when appropriate but explant as well. Overall, it should be clear throughout which assessments were performed when and which were done with BCI control, with BCI in place but without control, and with BCI removed (either because it hadn't been implanted yet or because it was explanted).

We appreciate this suggestion. We have added the sentence fragment (“in the native arm when BCI control was not in use”) at the location requested. In addition, we have added text throughout that “Motor Outcomes” paragraph; in fact, to make it as explicit as possible, the section has been retitled, “Unassisted Motor Outcomes.”

(2) Zheng "Trial of Contralateral Seventh Cervical Nerve Transfer for Spastic Arm Paralysis" should be cited and parallels drawn to given that this is essentially a lesion bypass approach.

We are familiar with this important work and are in full agreement that it should be cited and we have done so and added text to clarify the parallels with our report; we are grateful to Reviewer 3 for highlighting that work.

(3) The paragraph "The improvement that chronic stroke patients may achieve with mass practice..." is a good discussion but this should be expanded. There are patients with severe hemiparesis after stroke who do not regain any function even with rehabilitation(which would be passive range of motion and stretching exercises). For this population in particular, lesion bypass BCI offers an opportunity to regain some native movement and could potentially offer a rehabilitative approach. The authors should discuss this as a future direction, making sure to keep the distinction between assistive and rehabilitative BCI clear and the overlap here.

We thank Reviewer #3 for this suggestion and have followed it by adding additional text. We agree it is important to clarify the distinction between “assistive” and “rehabilitative” BCI. Overall, we wish to

express gratitude to Reviewer #3 for well-considered feedback that we think has made the manuscript more balanced and coherent, and in doing so also has helped us all honor the contributions of the participant, and his supportive family, in their altruistic motivation to advance research with the hope of helping other people who have had to navigate life with the limitations imposed by chronic stroke.

Respectfully,

Mijail Serruya, Ph.D., M.D.
Assistant Professor of Neurology
Thomas Jefferson University